# Dietary Carotenoids in Head and Neck Cancer—Molecular and Clinical Implications

**DOI:** 10.3390/nu14030531

**Published:** 2022-01-26

**Authors:** Katarzyna Starska-Kowarska

**Affiliations:** 1Department of Physiology, Pathophysiology and Clinical Immunology, Department of Clinical Physiology, Medical University of Lodz, Żeligowskiego 7/9, 90-752 Lodz, Poland; katarzyna.starska@umed.lodz.pl; Tel.: +48-604-541-412; 2Department of Otorhinolaryngology, EnelMed Center Expert, Lodz, Drewnowska 58, 91-001 Lodz, Poland

**Keywords:** head and neck cancer (HNC), carotenoids (CTDs), cellular signalling, cell cycle progression, apoptosis, chemoprevention, invasion, nutritional supplement, oxidative stress, anticancer therapy

## Abstract

Head and neck cancer (HNC) is one of the most common cancers in the world according to GLOBCAN. In 2018, it was reported that HNC accounts for approximately 3% of all human cancers (51,540 new cases) and is the cause of nearly 1.5% of all cancer deaths (10,030 deaths). Despite great advances in treatment, HNC is indicated as a leading cause of death worldwide. In addition to having a positive impact on general health, a diet rich in carotenoids can regulate stages in the course of carcinogenesis; indeed, strong epidemiological associations exist between dietary carotenoids and HNS, and it is presumed that diets with carotenoids can even reduce cancer risk. They have also been proposed as potential chemotherapeutic agents and substances used in chemoprevention of HNC. The present review discusses the links between dietary carotenoids and HNC. It examines the prospective anticancer effect of dietary carotenoids against intracellular cell signalling and mechanisms, oxidative stress regulation, as well as their impact on apoptosis, cell cycle progression, cell proliferation, angiogenesis, metastasis, and chemoprevention; it also provides an overview of the limited preclinical and clinical research published in this arena. Recent epidemiological, key opinion-forming systematic reviews, cross-sectional, longitudinal, prospective, and interventional studies based on in vitro and animal models of HNC also indicate that high carotenoid content obtained from daily supplementation has positive effects on the initiation, promotion, and progression of HNC. This article presents these results according to their increasing clinical credibility.

## 1. Introduction

Head and neck cancer (HNC) constitutes a group of cancers deriving from the mucosa of the upper aerodigestive system, including the oral cavity, nasopharynx, oropharynx, hypopharynx, and larynx, the latter of which represents the most common histological subtype [1,2]. Most head and neck cancers are squamous cell carcinomas (HNSCC). Although malignant head and neck tumours arise from the squamous epithelium, all forms possess biologically and clinically distinct features which determine their individual risk parameters, molecular intranuclear factors, chemoprevention effects, cure results, and long-term prediction [3]. According to Global Cancer Statistics (GLOBCAN), HNSCC was the sixth most common type of cancer worldwide. In 2018, it was reported that HNC accounts for approximately 3% of all human cancers (51,540 new cases) and is the cause of nearly 1.5% of all cancer deaths (10,030 deaths); this incidence had risen to 870,000 cases by 2020, accounting for 4.5% of human malignant cancers [4,5]. It is anticipated to rise by a further 30% to about one million new cases annually by 2030 (GLOBOCAN; gco.iarc.fr/today) [4,5]. These alarming patient mortality rates, characterised by 5-year overall survival rates under 50%, have been attributed to a high incidence of locoregional recurrences and/or local and distant metastases [6]. Despite the availability of increasingly accurate molecular and genetic diagnostic methods and tests indicating the histological progression of the tumour from cellular atypia through dysplasia and in situ cancer to invasive form of HNSCC, most patients are diagnosed with cancer in a very advanced clinical stage [7]. On the other hand, despite the development of cancer in a diagnostically accessible and visible anatomical region and the fact that HNSCC is usually preceded by the presence of precancerous lesions in the oral cavity, such as leukoplakia, erythroplakia, and oral submucosal fibrosis, most cases of HNSCC are diagnosed too late, resulting in poor treatment outcomes and high mortality.

The most important risk factors of HNSCC are carcinogens linked to smoking and alcohol abuse, both of which have been found to have considerable impact in various populations in epidemiological studies. It has been found that 10-year former smokers demonstrate half the ORs for oropharyngeal malignant tumours and a one-third to four-fifths reduction for cancer of the larynx, compared with current smokers; in addition, the OR of getting HNSCC is approximately 2.13-times higher in active smokers when compared with never smokers. Similarly, studies have also noted a quantitatively dependent relationship between alcohol drinking and the development of HNSCC; however, this association was found to be stronger among oropharyngeal and hypopharyngeal tumours than those of the oral cavity and larynx [8,9,10].

It is also important to note that the carcinogenesis of oropharyngeal HNS is closely related to infection with oncogenic strains of the human papillomavirus (HPV), mainly HPV-16 and HPV-18 [11,12]. HPV-associated HNSCC is a molecularly and clinically distinct subtype that is more typically in young subjects in developed countries and that is associated with better clinical outcomes [8]. In recent years, clinical trials have indicated that the presence or absence of HPV infection is an individual prognostic indicator for oropharyngeal cancer. In particular, the researchers have observed that second primary tumours and scattered metastases have significantly lower frequency in HPV-positive patients than HPV-negative subjects; this explains the correspondingly longer overall survival time in patients with HPV-related tumours (82.4% at three years) and a nearly 58% mortality decrease compared with HPV-positive patients (57.1% at three years) [11,12]. Furthermore, it has been found that subjects with HPV(+) oropharyngeal tumours show better clinical results to chemoradiotherapy and thus are eligible for new strategies of de-escalation treatment to reduce treatment-related toxicities and morbidity [6,11,12]. Moreover, the pathological tumour–node–metastasis system (pTNM) of HNC has been completed by the new 2017 8th Edition AJCC/UICC staging system, which contained characteristics relevant to HPV(+) tumours [7,13]. There is hence a necessity for better cognizance of the effect of CTDs on HPV-dependent HNCs, as these tumours have a unique biology which directly influences the effect of carotenoids, including chemopreventive ones, and may be of great clinical importance.

A very important source of the latest epidemiological data on HNC is information provided by researchers from the International Head and Neck Cancer Epidemiology (INHANCE) Consortium. A body of recent INHANCE-based publications—reviewed in 2015 and comprising over 35 studies encompassing around 25,000 cases and 37,000 controls from the United States, Europe, South America, and Asia—indicates that nearly 30% of patients have recurrent disease and 9% second primary cancers, with low overall and disease-specific 5-year survival rates of 51% and 57%, respectively [8,14]. The activities of the INHANCE Consortium is based on the international cooperation of a large number of research scientists which has performed a series of large epidemiological case–control studies of HNC (https://www.inhance.utah.edu/ (accessed on 28 May 2020). A large group of cases and controls provides an opportunity for reliable exploration of the associations between diet/nutrition and HNC initiation and development, while avoiding the residual confounding from smoking. For instance, a large sample-size study based on the INHANCE consortium found strong inverse relationships between fruit and vegetable intake, these being rich in CTDs, and HNC risk [8,14,15]. This study mainly contained American and European data. Other studies carried out in Taiwan and Brazil have also found fruit and vegetable consumption to be related to a decrease in HIV risk [16,17]. Despite this, it is important to highlight that many preclinical and clinical research studies of CTD supplements did not confirm a relationship between individual nutrients and the development of HNSCC. This may indicate that one nutrient cannot have an equivalent synergistic effect of many ingredients showing chemopreventive properties present in whole foods and in diets rich in fruits and vegetables, which are important sources of CTDs [18,19].

Convincing research has also demonstrated that diet and nutrition has significant importance in the carcinogenesis and development of numerous chronic affections and conditions, not only in diabetes, cardiovascular diseases, obesity, and osteoporosis but also in HNC region [11,20,21,22]. For instance, The Third Expert Report of the WCRF and the AICR has published unambiguous conclusions of an essential association between nutrition and the risk, carcinogenesis, and development of a range of cancers, including HNC and cancer of the stomach, lung, liver, kidney, breast, and prostate [11,22]. However, it is nevertheless important to note that evidence of dietary effects on the probability of development of human cancers of various origin remains relatively poorly defined due to the large diversity of head and neck region localization and the period being difficult to estimate between the exposure and its impact, i.e., tumour initiation and progressive growth [11]. On the other hand, WRCF and AICR have recently presented strong evidence showing statistically valid and reasonable relationships between diet and nutrition—particularly based around non-starchy vegetables, fruits, and foods rich in CTDs—and the onset of various cancers, including HNSCC, stomach, lung, kidney, breast, and prostate cancer; they have also been associated with carcinogenic tumour cell evolution and progressive steps of cancer progression [10,11,22].

Recently published works have examined the diverse diets consumed in different countries and inhabitants around the world. Such differences have supplied interesting information, pointing to an inverse relationship between the eating of fruits and/or vegetables and the development of head and neck carcinoma (HNC). For instance, subjects who report eating food lacking in fruits and/or vegetables, including foods with a scarcity of carotenoids, have twice the risk of HNSCC carcinogenesis when compared with patients who reported regular daily consumption [11,16]. Similarly, a pooled analysis of a few thousand patients with HNC—i.e., cancers of the oral cavity, oropharynx, hypopharynx, and larynx—and healthy controls from the International Head and Neck Cancer Epidemiology (INHANCE) consortium also noted that higher vegetable and/or fruit consumption was related to a significantly reduced risk of cancer. Interestingly, the researchers indicated that subjects with both low CTD consumption and high smoking status and alcohol abuse were at substantially higher risk of HNC [11,14]. This may be due to the fact that carotenoids can affect either cancer onset or further progression [11,23]. Recent studies have also found BMI and comorbidities linked to obesity to be associated with HNSCC progression [24,25]. Interestingly, combined consumption of CTDs and vitamin C and E can decrease the development and growth of pharyngeal and/or laryngeal cancer. Moreover, dietary carotenoids appear to bestow a stronger protective effect on individuals reporting greater alcohol consumption [8]. However, it should also be emphasized that a few publications do not confirm these observations, indicating that the dietary intake of various carotenoids or their serum concentrations do not appear to be associated with the onset, progression, and metastases in cancers of various origin [26,27,28,29,30,31].

Fruits and vegetables include substances having a biological effect, not only carotenoids, which can influence the subsequent phases of initiation, carcinogenesis, and cancer progression, as well as chemotherapy results [32]. Phytochemicals such as dietary carotenoids have been extensively explored as antioxidants, anti-inflammatories, immunoprotectors, immune regulators, cellular membrane stabilizers, and oncogenic signalling and apoptosis regulators, as well as cell-cycle and angiogenesis valid controllers [22,23]. In particular, the scavenging ability of carotenoids reduces reactive oxygen species (ROS), promotes DNA repair, negatively regulates oncogenic transcription, and stimulates the transcription of a number of key genes encoding antioxidant enzymes [33,34]. Interestingly, except in the presence of unbalanced cellular redox reactions and at high oxidation pressures, an elevated level of carotenoids may induce prooxidant reactions [34,35]. Moreover, inflammation has been associated with higher probability of recurrence and death due to head and neck cancer. It has been proposed that DNA methylation signatures in leukocytes may reflect systemic inflammation in HNC; in this regard, epigenetic stratification of HNC survivors has been observed to display various levels of lycopene concentration and alcohol dependence, as well as epigenetic changes (DNA methylation) in immune regulatory genes [36,37]. Recent publications comparing circulating carotenoid and cytokine levels with quantified factors such as alcohol abuse and tobacco dependence also suggest an association between DNA methylation and inflammation [36,37].

Recent years have seen considerable progress in identifying the carotenoid-based cellular and molecular regulatory mechanisms governing the promotion, development, and progression of cancers of various origin [23,33,35]. The potent antiproliferative effects, productive tumour inhibition, and antimetastasis properties of carotenoids against various cancer cells are believed to be stimulated by a large array of intracellular pathways; these are believed to act by modulating the function of key regulators, including nuclear-factor NF-κB, PI3K/AKT, ERK, p38, and c-Jun, JNK, and MAPKs, as well as various other cellular signalling proteins and transcription factors [35]. It has also been proposed that carotenoids may also exert their anticancer properties by blocking the PI3K/AKT/mTOR pathway and MYC family, Wnt, and Notch and by regulating the cell cycle progression in certain phases; they may also slow proliferation and angiogenesis by influencing gap junction intercellular communication (GJIC) and multidrug resistance (MDR) [23,35].

Nevertheless, in spite of the introduction of new, progressive methods of treatment into clinical practice, the 5-year survival of HNC does not exceed 50%, which may be due to the high tendency of these tumours to locoregional recurrence and MDR phenomenon. Interestingly, each of the carcinogenesis steps is known to be affected by dietary chemicals, including carotenoids. CTDs, as natural compounds present in food and beverages, are known for their intracellular antioxidant activities [38]. Several plant-derived phytochemicals, such as carotenoids, prevent and delay the primary carcinogenesis steps and have been found to decrease the risk and/or local progression and expansion of carcinogen-induced tumours in head and neck oncogenesis models [38].

This paper summarizes the expression and function of dietary carotenoids in HNC. It also discusses the contradictory roles of dietary components acting as sources of carotenoids, the relevant and up-to-date scientific data, the possibility of future carotenoids use in clinical practice, and the prospects of these phytochemicals regarding the risk, development, and aggressiveness of HNC neoplastic changes based on the latest literature.

## 2. Material and Methods

To demonstrate exhaustive data about the role of dietary carotenoids in the promotion, development, prevention, i.e., chemopreventive effects, and potential therapy of head and neck cancer (HNC) of various site origins, the corpus of studies encompasses a wide range of molecular, observational, and intervention studies. The review discusses scientific publications published between January 2001 and September 2021, all of which are accessible via the PubMed/Medline database. The following keywords were used: “carotenoids”, “nutrient”, “nutrients”, “dietary”, “diet”, “antioxidant”, “head and neck cancer”, “head and neck neoplasms”, and “head and neck carcinoma”. There was no restriction on language or research group characteristics. No exclusion criteria were employed.

This work shows the results of studies followed by latest preclinical works, i.e., animal and in vitro models of HNC, and clinically the most important case–control, cross-sectional, long-term observational, and intervention population studies and key opinion-forming systematic reviews from the International Head and Neck Cancer Epidemiology (INHANCE) Consortium. The scientific observations and conclusions are analysed in order of increasing clinical relevance.

## 3. Results

### 3.1. Chemistry of Dietary Carotenoids and Their Main Food Sources and Bioavailability

The carotenoids comprise a very broad group of tetraterpenoids, which are natural organic C_40_ pigments produced by plants and microorganisms; they are known to be present at particularly high levels in yellow to red food. The class covers a range of fat-soluble particles, including various hydrocarbons (carotenes) and their oxygenated derivatives (oxycarotenoids or xanthophylls) [39,40]. A classification of the major representatives of the carotenoids, viz. carotenoids, xanthophylls, and apocarotenoids, is presented in Figure 1.

Their structure is based around eight isoprenoid units with conjugated double bonds. The reverse placement of the units at the centre of the particle creates a system with the two central methyl groups being in a 1.6-positional connection and the nonterminal ~CH_3_ groups in a 1.5 position [39,40]. In the xanthophylls, the oxygen atom is replaced with a hydroxyl, epoxide, carbonyl, or one of many other functional groups [41,42,43]. All photosynthetic organisms (photoautotrophs), including some non-photosynthetic bacteria, macroalgae, insects, plants, and fungi, contain carotenoid pigments that act as auxiliary pigments in photosynthesis; the molecules also secure these organisms against chlorophyll auto photosensitization [39,41,44]. As most animals cannot synthesize carotenoids, they must supplement their diet with these pigments [41,45].

Although 700 carotenoids have been extracted, identified, and characterized in detail, only about 30 to 50 are metabolic precursors of vitamin A (all-*trans*-retinol, ROL) and have vitamin A activity. These are referred to as provitamin A carotenoids [39,41]. β-Carotene is converted to retinol more efficiently than other provitamin carotenoids, and it has 100% vitamin A activity. Other than β-carotene, the provitamin A β-carotenoids include α-carotene (which constitutes 50–54% of vitamin A activity), γ-carotene (42–50%), β-zeacarotene (20–40%), cryptoxanthin (50–60%), β-apo-8′-carotenal (72%), and β-apo-12′-carotenal; the latter is a molecule with high activity (~120%) similar to that of β-carotene [39,41]. Vitamin A activity requires the presence of an unaltered β-ionone ring with an attached polyene side chain containing eleven carbon atoms in the carotenoid molecule. Shortening the molecule by removing at least one end beyond a designated location, as in apocarotenoid molecules, reduces vitamin A activity. In contrast, the non-provitamin A CTDs include lycopene, lutein, canthaxanthin, zeaxanthin, astaxanthin, crocetin, and capsanthin [39,41]. Apocarotenoids and xanthophylls show reduced or even lack of vitamin A potential; however, many still have significant antioxidant activity [39,41,46,47].

The transformation of various provitamin A CTDs to vitamin A occurs with the participation of two converting enzymes: β-carotene-15,15′-oxygenase (BCO1) and β-carotene-9′-10′-oxygenase (BCO2). The action of BCO1 produces retinoids (β-15′-apo-carotenoids, C_20_) through symmetric division. Interestingly, BCO2 enzyme also can metabolize non-provitamin A composites such as xanthophylls through eccentric cleavage, thus promoting the formation of long-chain β-apo-carotenoids (>C_20_) [48,49,50,51]. The physiological action related to BCO1 and BCO2 function are determined by their location in various intracellular compartments: BCO1 is present in the cell cytoplasm, while BCO2 is inherent in the mitochondria [49,50]. The different cellular localization decides their physiological functions. In human cells, β-carotene is transformed by BCO1 intracellular, while β-cryptoxanthin and other asymmetric carotenoids undergo changes in mitochondria, i.e., BCO2 converts β-cryptoxanthin by oxidative cleavage at the C_9_, C_10_ double bond yielding β-apo-10′-carotenal and 3-hydroxy-β-ionone. β-Apo-10′-carotenal is transformed into alcohol and carried to the cytoplasm by unidentified specific proteins. β-Apo-10′-carotenol undergoes esterification by LRAT (retinol acyltransferase) and/or conversion by BCO1 by oxidative cleavage to all-trans-retinal [48]. Most importantly, β-carotene-15,15′-oxygenase has six common single-nucleotide polymorphisms (SNPs) responsible for the formation of partly or completely ineffective BCO1 forms [39].

Despite the fact that nearly 700 carotenoids are known, only 24 have been observed in blood and human cells, and 2 of them are detected in the eye tissues. Based on a broad review of relevant studies, the WHO has recently confirmed that a healthy fruits and vegetable diet has significant importance in the prophylaxis of cancer initiation and progression. They also identify a number of edible plant metabolites as components with biological activity and possible features that prevent carcinogenesis, inter alia, CTDs.

By dint of their proven health-protecting functions, including their positive influence on photoprotection, cardiovascular protection, immunoprotection, anti-aging, cellular membrane stabilization, and oxidative stress modulation, the consumption of CTDs has increased significantly in recent years. A considerable body of recent in vitro and in vivo studies has proved the notable results of CTDs in decreasing oncogenesis and in inhibiting its local and generalized extension; they have been associated with regulating oxidation, phosphorylation, and intracellular kinase signalling, programmed cell death, cell cycle process, gap junction intercellular communication (GJIC), angiogenesis, tumour cell spreading, and multidrug resistance (MDR) [41,52,53,54,55,56,57,58,59]. These studies provide sufficient evidence that dietary carotenoids bestow a protective effect against cancer-caused mortality. A few case–control, cohort, and randomized studies have also confirmed a negative relationship between CTD food intake and the occurrence of a variety of types of tumours, including breast, colorectal, lung, and prostate cancers, but also tumours in the head and neck region [41,60,61,62,63]. It should also be emphasized that not all publications confirm these observations and that some do not indicate any significant association between food intake or plasma levels of CTDs and the onset, progression, or metastases of various cancers; for example, lycopene consumption and serum concentration was not found to be related to the probability of development of gastric cancer, colorectal cancer, or ovarian cancer or even oral leukoplakia and oral cancer [26,27,28,29,30,31]. However, it should be noted that the malignant tumours specified in the above overview of carotenoid properties are of a completely different origin to those discussed in the present study. In addition, it is not easy to indicate unambiguous result interpretations based on their findings, as the studies use a wide variety of research methods and a diverse range of materials varying in tumour type and origin (leukoplakia, carcinogenesis, neoplastic metastasis); considerable differences can also be observed among such varied HNCs (laryngeal carcinoma, oropharynx carcinoma, nasopharyngeal carcinoma, salivary carcinomas) with different histological differentiation status and proliferative index. Furthermore, most studies on carotenoid alternations concern various types of carotenoids circulating in the bloodstream or present in food. It should also be borne in mind that one dietary carotenoid may not recreate the synergistic action of the numerous chemoprotective compounds present in whole foods and may not yield the same results.

CTD particles have structure of long polyene chain of 8–13 conjugated double bonds. This type of construction enables effective prevention of reactive oxygen–nitrogen species formation (ROS, RNS) [41]. Importantly, three main mechanisms for inactivating anti-ROS and anti-RNS activity can be identified: ē transmission between peroxyl radicals (ROO)• and CTDs, which leads to creation of the carotenoid radical cations (CAR+•) or radical anions (CAR−•); radical adduct production (ROO-CAR)•; and H atom transmission, allowing the formation a neutral CAR radical (CAR)• [41,64]. Occurrence in the molecule of the hydroxyl, epoxide, or carbonyl functional groups, among others, determines the functional properties of the carotenoid, such as its anticancer potential and cancer-related molecular events [41,65].

In addition to the well-known antioxidant properties of carotenoids, which can form the basis of cellular antitumour mechanisms, i.e., chemopreventive effect, inhibiting tumour onset and progression, it is also necessary to note that carotenoids can also have pro-oxidative activity under certain conditions, such as uncontrolled cellular oxidation state, increased pO_2_ in blood, and higher CTD level. At low oxygen pressures, carotenoids can function as potent antioxidants. At high oxygen plasma levels, CTDs may act as readily autoxidized molecules, thus exhibiting prooxidant activities [41,46,66,67]. Most importantly, tumour cells in native status are characterized by increased intracellular levels of ROS compared with normal cells. This phenomenon has serious intracellular consequences, i.e., under the increased levels of ROS in the tumour intracellular environment, the pro-oxidant function of CTDs dominate over their antioxidant characteristics and stay related to elevated status of oxidation; this can entail various intracellular effects such as tumour cell development, proliferation, and apoptosis of cancer cells [41,68,69,70]. Moreover, it has been noted that carotenoids may also elevate intracellular hydrogen peroxide levels in cancer cells while normalizing H_2_O_2_ levels in normal cells [41,56]. The last two decades have seen many scientific publications that have shown that the enhanced accumulation of ROS by CTDs in cancer cells constitute a key mechanism of selective killing of target cancer cells. In addition, they have been observed to remarkably reduce the cytotoxic influence of ROS-generating antitumor drugs (e.g., doxorubicin) in noncancerous cells due to their antioxidant properties, while demonstrating potent synergetic pro-oxidant activity in the killing of tumour cells [71,72].

From nearly 40 nutrient CTDs, only a few molecules of them, viz. lycopene, β-carotene, α-carotene, lutein, zeaxanthin, and β-cryptoxanthin, constitute >95% of the total CTDs in the plasma [66]. Importantly, their concentration in tissues and blood are known to be influenced by the geographical region in the world, BMI, sex, smoking habit, and alcohol consumption. The total CTD consumption in European populations is on average a value of ∼9.5–16 mg/d and mainly concerns β-carotene (∼3–6 mg/d) [73]; in addition, in the Europeans, the mean blood concentration of lycopene was estimated to be an average value of 0.43–1.32 μM, lutein (0.26–0.70 μM), β-carotene (0.21–0.68 μM), β-cryptoxanthin (0.11–0.52 μM), α-carotene (0.06–0.32 μM), and zeaxanthin (0.05–0.13 μM) [66]. Importantly, carotenoids are characterized by relatively low bioavailability. This may be due to the fact that carotenoids, being lipophilic molecules, are not easily released into the GI; in addition, their solubility in the gastrointestinal fluids is also very low, but this may be improved by the creation of micelles, formed by bile salts, phospholipids, and lipid digestion metabolites. As they are naturally trapped in specific vegetable or fruit (dietary) structures, homogenization could also improve their release from food. In addition, dietary carotenoids may also interact with soluble food fibre and other digested molecules, such as plant derived sterols, which may also limit their bioavailability [41,74]; for example, the bioavailability of lipid-soluble lycopene is greatly increased in fatty food [28]. Interestingly, lycopene present in pasteurized and conserved tomato products is frequently higher than in natural tomatoes because the thermal/chemical reactions cause water loss. Examples of the major dietary carotenoids, their chemical structure, and food sources are illustrated in the Figure 2 [39,66,75].

The last two decades, have seen a great deal of research on the physiological roles and pharmacological activity of carotenoids. Importantly, CTD level in plasma is recognized by many researchers as an effective indicator of dietary intake [39,76]. For example, healthy people who have consumed a low-carotenoid diet (<0.4 mg/d) for two weeks presented plasma carotenoid levels that were <60% of the initial concentrations noted at the beginning of the experiment; however, these levels increased to original values after the subjects ate vegetables and fruits rich in CTDs for one week [39]. Hence, carotenoid appears to be a reliable reference parameter of CTD consumption.

In humans, dietary carotenoids are stored in adipose tissue and hepatocytes, as well as ovaries and testes, with the highest levels being observed in adrenal glands, testes, and ovary tissues [39,76]. The estimated carotenoid level in blood plasma usually reflects the dietary concentrations in total serum, typically with concentration of 20–25% of the total carotenoids in human plasma [39]. Interestingly, lycopene (ΨΨ-carotene; C_40_H_56_), a noncyclic red lipophilic carotenoid found in tomatoes and other red or pink-orange fruits, has recently aroused great scientific attention; it is believed to be the most predominant carotenoid in human plasma, and it constitutes more than 50% of the carotenoid content in human serum. Lycopene does not exhibit the characteristics of provitamin A because it does not have a β-ionic structure in its molecule [28]. Furthermore, although the predominant isomer in dietary sources is the *trans*-form, 50% of the total lycopene in human plasma is in the form of *cis*-isomers (i.e., 5-, 9-, 13-, 15-, 7-, and 11-) [28,77]. While the main action of lycopene is to assimilate the sun’s rays during the course of photosynthesis, it also protects plants against photosensitization. Studies indicate that lycopene is the best-known scavenger of ^1^O_2_ and free oxygen radicals of all the carotenoids and that it has the ability to reduce oxidative stress and prevent reactive oxygen species (ROS) generation [28], thus protecting cells against the oxidation of intracellular molecules and deoxyribonucleic acids [78,79].

A large number of epidemiological studies and clinical trials have highlighted the advantages and protective action of natural tomatoes and processed vegetables containing ΨΨ-carotene as antioxidant, anti-inflammatory, and anticarcinogenic factors. They also appear to not only decrease the risk of neoplasms but also significantly lower mortality from cancers in all sites [28,78]. For instance, recent studies by Rowels and collaborators [80,81] observe that consumption of an additional amount of 2 mg of dietary lycopene is related to lower risk of prostate cancer development by 1% and that prostate carcinogenesis is lower by 3.5–3.6% in patients with an increase in the level of serum by an additional 10 μg/dL. However, a lot of works have found that ΨΨ-carotene consumption and increased plasma level are not associated with any such benefits, and furthermore may in fact favour the promotion, progression, and spread of gastrointestinal, colorectal, and ovarian cancers, among others [26,27,28,29,30,31].

Interestingly, research indicates that increasing vegetable and fruit intake, these being rich sources of CTDs, can diminish cases of either oral leukoplakia HNC or weaken the intensity of radiotherapy-induced side effects and local recurrence rates for HNC patients. However, due to the location of the tumours and the swallowing disorders resulting as the side effects of radiotherapy, patients with HNC often eat small quantities of nutrients in the diet and consequently could acquire low levels of carotenoids in the blood plasma [76,77]. Indeed, studies have found the serum of patients with HNC to contain significantly lower levels of β-carotene and lycopene, as well as lutein, zeaxanthin, α-carotene, and total carotenoids, vs. healthy control individuals. The mean β-carotene food intake of HNC cases can even be half that in the control group [82]. It should also be added that the reduction in mean food intake of β-carotene observed during RTG therapy appeared to be accompanied by an increase in the risk of heavy undesirable effects and local recurrence. It is also known that the use of carotene supplements and high doses of carotene in food effectively prevent the occurrence of radiation therapy side effects; this has been attributed to β-carotene bestowing protective effects against radiation injury on noncancerous tissues. Nevertheless, it is important to note that while carotene supplements can increase cancer recurrence rates, no such problem has been observed for dietary β-carotene. In addition, carotenoid levels were found to be three to four times lower in HNSCC patients after receiving radiotherapy than in healthy controls. Similarly, carotenoid concentrations in the blood of the HNSCC patients’ postradiation treatment was favourably associated with progression-free rate and survival [82].

In addition to having antioxidant abilities and ^1^O_2_ capture capabilities, CTDs are also known to participate in the regulation of gap junction intercellular communication (GJIC). Carotene deficiency has been found to be significantly related to cancer intracellular cell pathway abnormalities and to uncontrolled cell proliferation and leukoplakia. The epidemiological and experimental data clearly indicate that carotenes and CTDs show significant activity in preventing the development of oral epithelial cancer, and an inverse relationship between serum carotene and retinol was also documented for this cancer risk [77].

### 3.2. The Molecular Role of Carotenoids in Mechanisms Responsible for Carcinogenesis

#### 3.2.1. Molecular Aspects of Head and Neck Carcinogenesis

Tobacco smoking and alcohol abuse are well known risk factors of HNC carcinogenesis. Unfortunately, the role of nutrition and dietary customs, job-related chemical exposure, and dietary habits/gene–environment interactions, remains inconclusive. Unfortunately, due to the heterogeneity in HNS tumour biology; differences in trials projects, protocols, and the size of the studied groups; a relatively small number of observational studies; and the overwhelming role of tobacco smoking and alcohol consumption, it is not possible to precisely estimate the relative contribution of other HNS risk factors. Further explanation of the molecular events involved in HNSCC development may help identify potentially effective biomarkers and provide new procedures for chemoprevention and targeted therapy. The pathogenesis of HNSCC is a multistep process involving the progressive accumulation of molecular alterations [11,83,84]. HPV(−) HNC carcinogenesis commonly engages the inhibition of tumour suppressor genes, such as *CDKN2A* (encoding the inhibitor p16^INK4A^) and *ARF* (encoding p14, a stabilizer of p53), as well as *TP53* in early stages and *PTEN*; it has also been associated with LOH (loss of heterozygosity) of the 6p, 8, 4q27, and 10q23 genes at later stages. The occurrence of key intracellular genetic changes, i.e., significant tumour suppressor genes mentioned above or numerous essential signalling pathways such as the AKT/PKB and MAPKs, is related to the development, progress, and poor outcome in HPV(−) HNC [11,84].

Tobacco-derived carcinogenesis has been associated with the elevated level of α-7 nAChRs receptors, which promote cell divisions and the formation of metastases via the formation of a phosphorylated form EGFR, AKT, mTOR, and the activation of β-sympathomimetic receptors. First, carcinogenic substances in tobacco smoke—i.e., nicotine, polyaromatic hydrocarbon (benzo-(a)-pyrene, nitrosamines, or acrolein, etc.)—increase the level of intermediate filaments (fibronectin and vimentin), whereas they decrease β-catenin–E-cadherin complex, thus favouring cell dissemination through induction of epithelial–mesenchymal transition (EMT). Nicotine is known to determine the effectiveness of anticancer therapeutic effects via cytochrome P450 (CYP)-mediated processes, and tobacco consumption has been found to promote a proinflammatory tumour microenvironment, further supporting tumour growth. Furthermore, alcohol can directly upregulate the expression of vimentin and matrix metalloproteinases MMP-2, MMP-7, and MMP-9, which are known to promote an epithelial–mesenchymal transition, cancer cell aggressiveness, and extracellular remodelling [11,85].

It is well known that HPV-positive cancers are more likely to have widespread genetic modifications linked to DNA repair processes. Recent research has also indicated that alterations in numerous cell signalling pathways related to activity of phosphoinositide 3-kinase (PI3K) may be connected with better survival rates in those patients [11,86,87]. The researchers indicate that the probable sources of HPV(+) HNC carcinogenesis are cells belonging to stem cells or proliferating basal cells. The HPV DNA joins into the host genome, and the viral oncoproteins disturb key intracellular signalling pathways responsible for cell cycle control; this is facilitated by the impaired function of the tumour suppressor molecules p53 by E6 and pRb via E7. Such activity promotes tumorigenesis, leading to invasive and metastatic phenotypes [11,83].

These important molecular and genetic mechanisms known to determine the development of head and neck cancers may be important targets for dietary or supplemental carotenoids. In precancerous and carcinogenic intracellular disturbances, these phytochemicals may contribute to inhibiting tumour cell divisions, cell cycle regulation, and pro-apoptotic effects and to decreasing angiogenesis; thus, their administration may inhibit the development, progression, and metastasis of tumour cells because of their potential antioxidant, anti-inflammatory, and immunoprotective action, cellular membrane stabilisation, and inhibition of oncogenic signalling pathways and apoptosis activators [22,23]. In particular, the scavenging ability of carotenoids in reducing ROS, singlet oxygen ^1^O_2_, and free oxygen radical levels supports DNA repair, while inhibiting oncogene transcription and stimulating of the numerous important genes coding antioxidant enzymes known to inhibit human carcinogenesis [33,34].

#### 3.2.2. Molecular Role of Carotenoids in Major Intracellular Oncogenic Signalling

Due to having their long polyene chain and the action of heterogenous functional groups, CTDs can directly or indirectly regulate important molecular and cellular oncogenic processes, inter alia, key oncogenic cellular signalling processes. They can also control cell cycle progression, leading to cell cycle arrest, induce apoptotic processes, mediate the redox balance, restrain neoplastic metastasis, impede angiogenesis, and affect GJIC and MDR phenomenon linked to the cytotoxic antitumoral results in cancer cells [41]. Moreover, many in vitro and in vivo trials have found CTDs to play significant roles in various oncogenic intracellular pathways, including oxidative stress modulation, immunoprotection, cellular membrane stabilization, formation of a phosphorylated active form and photoprotection, regulation of key signalling kinases, apoptosis regulation, cell cycle progression, angiogenesis regulation, and cancer metastasis protection [41,52,54,56].

Integration of dietary CTDs and changes in intracellular signals are responsible for essential key reactions related to carcinogenesis, such as tumour cell proliferation and, consequently, tumour progression. The most widely studied cancer signalling pathways include PI3K, AKT/PKB, and mTOR; these may significantly affect the increased transcription process connected with cellular proliferation, distant therapy results, and contribute to chemosensitivity [88,89,90]. CTDs can modify crucial regulatory oncogenic pathways favouring the development of cancer, including head and neck cancers (HNSCC) [88,91].

Among numerous oncogenic tumour transcription systems, nuclear factor-kappa B/inhibitor of kappa B complex regulates the key tumorigenesis processes, i.e., the activity of oncogenic genes, cytokine generation, and cell divisions. The NF-κB signalling pathway is activated by phosphorylation, which is mediated by IκB kinase (IKK) and progressive proteasomal ubiquitination of IκB; this nuclear translocation of NF-κB subsequently activates oncogenic genes [88,92].

Similarly, AKT signalling regulates tumorigenesis by direct phosphorylation of mTOR, thereby inducing anti-apoptotic pathways [88,89,90]. For instance, a decrease in PI3K/AKT/mTOR system activity by CTDs blocks the activation of crucial oncogenic molecules, i.e., MYC, which are inhibited in almost 50% of human cancers, including head and neck cancer, small-cell lung cancer, neuroblastoma, and breast cancer [88,93]. This phenomenon may form the basis of potential anticancer therapies in MYC-dependent tumours. For example, astaxanthin treatments may suppress the mobility and dissemination of cancer tissue by a decrease in MYC transcription factor mediation [89,94]. Moreover, treatment with dietary CTDs such as astaxanthin, α-carotene, lycopene, and β-carotene may significantly decrease the status of the phosphorylation of AKT, JNK, MAPKs, and ERK1/2, thus reducing the tumour cell divisions and their life [88,92,95]; this family of mitogen-activated protein kinases retrieves a key role in the function of NF-κB and other intracellular oncogenic signalling, thus enhancing the survival of cancer cells. Carotenoids such as zeaxanthin may also negatively modulate ERK1/2 kinases by inhibiting the Ras–Raf–MEK–ERK signalling cascade, enhancing tumour cell survival and invasion [88,96].

#### 3.2.3. The Molecular Function of Carotenoids in Apoptotic Signalling and Cell Cycle Progression

Apoptosis is a precisely controlled phenomenon of programmed cell death. In numerous tumours, as well as HNC, essential pathological disturbances of mechanisms take place, resulting in the dysregulation of apoptosis. This may be related to the overexpression of key anti-apoptotic proteins such as Bcl-2 and Bcl-xL or inhibition of pro-apoptotic proteins such as Bcl-2/Bak1, Bad, and Bax [41,97,98,99]. Programmed cell death is determined by an intrinsic or extrinsic process. In the intrinsic path, the pro-apoptotic complex Bax/Bak1 facilitates the transition to the cytoplasm of both cytochrome c and Smac or Diablo molecules through the mitochondrial membrane. This phenomenon is connected with the creation of an apoptosome, whose key component is Apaf-1 factor; this stimulates the caspase-9/-3 cascade, which directly leads to apoptosis. Two other key inhibitors of programmed cell death are survivin and XIAP, which inhibit apoptosis by forming a complex with caspases. It has also been found that adding an acetyl functional group to p53 protein at Lys-120 causes an increase in its activity toward Apaf-1, and thus stimulates mitochondrial caspases. While the phosphorylated form of Bad allows Bcl-2 to restrain Bax/Bak-activated cell death, the dephosphorylated Bad protein inactivates inhibitors of apoptosis, i.e., Bcl-2 and Bcl-xL [41,97,98,99,100].

The extrinsic way process begins outside an intracellular matrix. At least two apoptotic systems can be indicated: one concerning death receptors (DRs, also called DR-5, Fas, TNFR-1) activity, and another is connected with cytotoxic stress. DRs constitute the TNFR superfamily, which is characterized by a cystein-rich extracellular part and a homologous intracellular part known as the death domain (DD). Intracellular molecules such as FADD, TRADD, or Daxx have DDs so that they can cooperate with the DRs and transfer the apoptotic signal in the cell. FasL (Fas ligand) functions as a ligand for Fas and leads to oligomerization of its receptor and then causes a collection of the DDs and activation of co-factor FADD.

After the FADD binding via its death effector domain (DED) motif to a homologous motif in procaspase-8, the death-inducing signalling complex (DISC) is activated. Upon recruitment by FADD, procaspase-8 splits and forms active caspase-8, which then activates caspase-3 and -7, leading to the cell to the programmed cell death [41,97,98,99]. A heterodimer formed by activated peroxisome proliferator-activated receptor gamma (PPARγ) and RXR increases the rate of apoptosis in a variety of cancer cells by binding to PPAR response elements and recruiting co-activators and various other nuclear regulatory proteins [41].

Administration of 8 μM β-carotene, astaxanthin, capsanthin, or bixin has been observed to reduce cell divisions of K562 tumour cells and to decrease their viability, induce the apoptosis process, and regulate cell cycle. These CTDs increased the activation of PPARγ and p21^Waf−1/Cip−1^ and downregulated the CCND1 gene. Both processes were dose- and time-dependent. In addition, β-carotene, astaxanthin, capsanthin, and bixin also may increase Nrf2, a crucial molecule of transcription in the Keap1-Nrf2/EpRE/ARE cellular signalisation [101]. In addition, in cells exposed to decreased levels of CTDs (i.e., 5 and 10 μM), astaxanthin was confirmed to most successfully inhibit proliferation, followed by bixin, β-carotene, and capsanthin. Furthermore, astaxanthin was observed to inhibit cell divisions in a hamster model of oral cancer via the regulation of cyclin D1 action, inhibiting JAK/STAT-3 kinases and proliferating cell nuclear antigen (PCNA) [102].

The carotenoid astaxanthin also plays important roles in the regulation of apoptosis. Purified astaxanthin at a typical dose of 5 μM or 25 μg/mL of astaxanthin-rich *H. pluvialis* extract can substantially raise the Bax, p53, p21^Waf−1/Cip−1^, and p27^Kip−1^ activity. In addition, astaxanthin treatments were also found to significantly decrease CCND1 translation and Bcl-2 level and AKT pathway activation [103]. Other studies have also indicated that astaxanthin appears to exert its antiproliferative, anti-apoptosis, and anti-invasion effects via different molecules and pathways including NF-κB, STAT3, and PPARγ factors. That is why astaxanthin constitutes a promising chemotherapeutic agent in malignant diseases [104].

Interestingly, many publications indicate that astaxanthin may have a dual nature regarding carcinogenesis. On the one hand, Satomi et al. [105] report that exposure of human cancer cells with astaxanthin led to a significant increase in levels of phosphorylation forms of MAPKs kinases family. These phosphorylated MAPK family proteins are believed to enhance tumorigenesis by activating transcriptional nuclear factor and other intracellular signals associated with tumour cell survival. This action is pro-oncogenic and intensifies the development and progression of tumour cells. On the other hand, astaxanthin also has strong anticancer properties associated with its effect on JNKs, which enhance the level of other key pro-apoptotic molecules such as GADD45A [41,105]. It has also been found that in vitro astaxanthin treatment significantly decreased the p65 unit of nuclear factor kappa B and COX-2 levels in dimethylhydrazine-induced rat colon carcinogenesis, while also reducing extracellular matrix production, i.e., matrix metalloproteinases-2 and -9 and inhibiting the activity of PCNA and ERK-2, and AKT kinases [106].

Another set of in vitro studies by Kavitha et al. [53] found that astaxanthin plays a significant role in relevant oncogenic intracellular pathways, including inhibition of NF-κB and Wnt/β-catenin pathways via inhibition of MAPKs kinases and PI3K/AKT/mTOR; this resulted in the induction of an intrinsic way of programmed cell death in a hamster model of oral cancer. In addition, treatment with astaxanthin resulted in the downregulation of Bcl-2, phosphorylation of Bad and higher survivin activity, and the stimulation of Smac/Diablo molecules and cytochrome-c, followed by their displacement into the cytosol; administration also resulted in the positive regulation of the PARP cleavage in hamster HNC carcinogenesis. The anti-inflammatory effect of this carotenoid is also important in the induction of apoptosis. Yasui et al. [107] have noted that dietary administration of astaxanthin (335 μM/kg diet) may considerably inhibit the cellular level of inflammatory molecules, including NF-κB, TNF-α, COX-2, and interleukin-1β, which also have significant anticancer activity.

Moreover, several works have highlighted the inhibitory function of β-related to tumorigenesis process. β-Carotene, at the normal achievable level of just 1 μM, efficiently decreased the activity of anti-apoptotic molecules such as Bcl-2 and PARP, as well as nuclear factor NF-κB, in human tumour MCF-7 cells [95]. β-Carotene experimental use was also found to influence AKT and ERK1/2 function and subsequent lowering in Bad phosphorylation. β-Carotene is able to inhibit the action of the antioxidant SOD-2, Nrf-2, and the endoplasmic XBP-1 activity [101].

Other CTDs, including astaxanthin, zeaxanthin, and fucoxanthin have also demonstrated considerable anticancer potential, e.g., fucoxanthin was noted to stimulate cytotoxicity and cell death by the intrinsic pathway in HL-60 human tumour cells [41,55]. A crucial part of this mechanism of action was the inhibition of proliferation by cell cycle arrest, which was enabled by stimulation of GADD45A, Fas/TNFR-1 DR, and caspase-3 expression and downregulation of Bcl-2 expression. This resulted in lower levels of Bcl-2 and Bcl-xL and a higher activation of Bak and Bax, cytochrome c, and caspase-3 and -9; in addition, transmission through mitochondrial membrane was increased.

Zeaxanthin has been suggested to act as a pro-apoptotic molecule, inducing cytotoxicity in human cell lines SP6.5 and C918 by inducing caspase-3 and caspase-7 activation and promoting PARP cleavage, as well as lowering Bcl-xL level [108]. Sheng et al. [109] found zeaxanthin to induce cell death by decreasing mitochondrial membrane potential; growing cytochrome c transmembrane permeation, Bax, cleaved-caspase-3 (cle-cas-3), and cleaved-PARP (cle-PARP) activation; and reducing Bcl-2, pro-caspase-3 (pro-cas-3), and pro-PARP expression. In addition, zeaxanthin was observed to inhibit cell cycle at the G2/M phase by increasing the levels of p21^Waf−1/Cip−1^ and p27 ^Kip−1^ and decreasing AKT, cyclin A, cyclin B1, and CDK1/2 action. Furthermore, zeaxanthin use was associated with increased expression of ROS, MAPKs kinase family phosphorylation, and nuclear factor kappa B inhibitor and with decreased level of phosphorylated form of ERK, AKT, STAT3, and nuclear factor kappa B.

Similar to fucoxanthin, β-cryptoxanthin has also demonstrated antiproliferative activity against human cancer by stimulating the initiation of the cell death intrinsic signal. Treatment results in inhibition of cell survival by interruption of the cell cycle in the G1 phase, the removal of a phosphate (PO_4_^3−^) group from the pro-apoptotic Bad, PARP cleavage via caspase-3 activity, increased flow of Ca^2+^ ions inside the cell, and a higher concentration of reactive oxygen and nitrogen species, which contribute to oxidative stress-induced apoptosis [110].

The most significant antitumour features of CTDs are realised by antiprogressive characteristics and inhibition of the cell cycle. This is achieved by inhibiting the key proteins in the cell division process at critical phases; a good example is the suppression of the phosphorylation of the tumour suppressive Rb molecule, which is an important factor regulating S phase transcription [41]. Almost all known dietary CTDs, such as β-carotene, lycopene, and fucoxanthin, may inhibit cell cycle progression and cell proliferation by various mechanisms. For instance, treatments of AtT20 cancerous human cells by β-carotene and lycopene were found to induce growth inhibition and pro-apoptotic effects by stopping the transition from the G phase to the G1 phase of the cell cycle, attenuating intercellular GJIC, growing level of phosphorylated connexin 43 and p27^Kip−1^, and reducing the expression of Skp2 protein; this has been found to have antitumour effects in neoplastic diseases of various origins [111].

Takeshima et al. [112] report that lycopene demonstrated inhibitory function against estrogenic/progestogenic receptors and the HER2 molecule in breast malignant cell lines by arresting cyclin D1 activity and upregulating the p21^Waf−1/Cip−1^ inhibitor, mainly by stopping the cell division process at the G0 and G1 points. Similarly, treatment with fucoxanthin inhibited cell cycle progression at the G0/G1 phase in B16F10 tumour cells and in the G2/M phase in cancerous MGC-803 cells. Interestingly, in the first case, fucoxanthin treatment induced apoptosis by the inhibition of Bcl-xL and IAPs at the protein level, resulting in sequential caspase-9 and -3 activity and PARP cleavage. It was also associated with decreased expression of phosphorylated-Rb (p-Rb), cyclin D1, and CDK4 molecules and increased level of p15^INK4B^ and p27^Kip−1^ [113]. In the second pathway, fucoxanthin blocked the JAK/STAT3 signalling-mediated expression of cyclin B1 and survivin [114]. Fucoxanthin inhibited the cell division cycle in GADD45A at the G2/M and G1 phases by interacting with Cdc2/CyclinB1 [105].

#### 3.2.4. The Molecular Role of Carotenoids in Angiogenesis and Cancer Metastasis

Angiogenesis is a pathophysiological phenomenon that is characterized by the formation of new blood vessels from preexisting endothelial cells. This is a key feature of solid tumours which allows accelerated growth and metastasis; as such, the suppression of angiogenesis may constitute an important potential treatment for many types of cancers. The process is characterised by high VEGF, FGF, and EGF factor activities. Due to their accelerated growth rate, many malignant tumours are subject to hypoxia, which promotes angiogenesis via induction of pro-angiogenic factors i.e., HIF-1α and angiogenesis-associated factors (AAFs) [115,116].

Most importantly, lycopene and apigenin are able to decrease angiogenesis by inhibiting new vessel formation; they also appear to inhibit endothelial cell migration. This is a process dependent on the action of endothelial growth medium (EGM)—a culture medium with growth factors such as VEGF and FGF [41]. The administration of 1.15 μM lycopene, representing physiological blood plasma levels, results in a considerable depletion in a net of new blood vessel branching and a decrease in the number of junctions, endothelial cells and tubule number, and new vessel length. Interestingly, lycopene also possesses a strongly marked inhibitory activity on angiogenesis that takes place as an independent process of downregulation of angiogenic factors such as VEGF and TNF-α. In contrast, lycopene also shows significant anti-angiogenic impact via stimulation of immune response and causes a crucial increase in IL-12 and IFN-γ concentrations and decrease in matrix metalloproteinase-2 (MMP-2) action in the tumour microenvironment [117,118,119,120].

Some studies have also suggested that fucoxanthin may possess potential anti-angiogenic properties by extinguishing the FGF-2 and FGFR-1 transcription, as well as EGR-1 expression in endothelial cells. Moreover, preclinical treatment with fucoxanthin was found to potentially downregulate the phosphorylation of ERK1/2 and AKT kinases stimulated by FGF-2 activity. As such, this carotenoid may be a potent direct repressor of the motility and cell division cycle of endothelial cells in various types of cancers [41,121].

However, few works discussing apoptotic activity of CTDs in HNC have been known. Researchers clearly indicate that CTDs exhibit a wide range of anticancer activities, including pro-apoptotic activity and stimulation of intracellular and extracellular signals of programmed cell death. A considerable body of preclinical oncology research indicates that the modulation/activation of apoptosis in HNC takes place at the level of translation process. Furthermore, CTDs have been found to show additive/synergistic effects if they co-work with standard oncostatic factors and to sensitize tumour cells to conventional anticancer treatment. Despite this, carotenoids dissolve poorly and thus have low bioavailability, and this is problem when treating solid tumours; however, this can be addressed using modern nanotechnological approaches, solid dispersions, microemulsions, or biofortification, which may be the lever to an effective design for personalized cure methods of HNC region tumours [41,122].

Another factor known to inhibit the process of angiogenesis is astaxanthin. It has been found that dietary astaxanthin may influence JAK-2/STAT-3 signalling in the 7,12-dimethylbenz-(a)-anthracene (DMBA)-induced hamster buccal pouch (HBP) carcinogenesis model. The data clearly indicate that dietary astaxanthin may prevent the development and progression of HBP carcinomas, a standard model of oral cancer, through the inhibition of key transcription signalling and its downstream events. Thus, astaxanthin appears to be a promising possible anticancer chemopreventive factor by inhibiting initiation and local/generalized aggressiveness of the tumour [102].

A number of recent studies indicate that CTDs may also intensively, and efficiently, control tumour invasion and metastasis, the complex phenomenon of acquiring adhesive features by malignant cells, progression, tissue infiltration, and lymphoid/blood vessel dissemination. This metastatic ability may be regulated by carotenoids on many levels, including the expression of important pro- and antimetastatic factors such as Nm23-H1; uPA; matrix metalloproteinases, e.g., MMP-1, -2, -3, -7, -8, -9, -12, and -13; MT1-MMP and MT3-MMP; endogenous tissue inhibitors of metalloproteinases (TIMPs), e.g., TIMP-1, -2, -3, and -4; transforming growth factor-β1 (TGF-β1); Rho subfamily, including RhoA, Rac1, and Cdc42; transmembrane E-cadherin; and surface glycoprotein, including CD44 and CXCR4, and HIF-1α. Carotenoids can also regulate pro-angiogenic factors, such as VEGF and cytokines or toll-like receptors (TLRs) [41,117,123,124,125]; these factors can contribute to the destruction and remodelling of the ECM compounds, which is a basic step in the further progression of the tumour.

It is worth mentioning here one of the few publications on the results of the comparative study of α-carotene and β-carotene and describing the potential impact of dietary CTDs on the aggressiveness and dissemination of solid tumour cells, including HNC. Treatment with 0.5–2.5 μM α-carotene for 48 h significantly inhibited the invasion, migration, and adhesion of human carcinoma model cells (SK-Hep-1) and other carcinogenic animal tumour models (PC-3, DU145, and LNCaP cells) compared with β-carotene [126,127]. Importantly, the action of α-carotene substantially impacted the effects of uPA, MMP-2, and -9 with a concordant greater activity of PAI-1, TIMP-1, TIMP-2, and Nm23-H1 molecules. Furthermore, α-carotene reduced PTK2/FAK, p38MAPK/ERK 1/2 kinases and ERK/p38/JNK signalling pathways, which induced activity of Rho/Rac 1 proteins [75].

In addition, β-carotene was found to inhibit the spread of tumour cells by dysregulation and lower expression of HIF-1α, VEGF, and GLUT1. In addition, β-carotene treatment resulted in significantly lower HIF-1α and GLUT1 concentrations both in vitro and in vivo [41,128]. β-Carotenes can also eliminate antimetastatic properties of tumour cells by inhibiting the translation and activation of MT2-MMP under both normoxia and hypoxia conditions [128]. Interestingly, application of β-carotene supplements contributed to a significant reduction in the occurrence of metastasis and tumour volume, with a concordant decrease in transcription of MMP-2 and increased activity of TIMPs (TIMP-1 and -2) in immunodeficient nude mice treated with SK-N-BE(2)C tumour cells, suggesting their independent association with MMPs. Alternatively, β-carotene may also exert its antimetastatic activity by inhibiting the Notch pathway and altering EMT, by increasing the activity of epithelial markers such as E-cadherin, ZO-1, and CK5, and by reducing the level of mesenchymal markers such as Snail-1, vimentin, and N-cadherin [75,129]. In addition, β-carotene was found to inhibit M2 macrophage polarization and fibroblast function via α-SMA, FAP, and TGF-β1; both types of cells modulate the behaviour of cancer cells in the tumour microenvironment. Application was also found to inhibit CSC markers, to modulate EMT markers, and to activate IL-6/STAT3 signalling, directly decreasing cancer cell invasiveness and migration [75,130].

Lycopene has a slightly different effect that inhibits the local and global development of the tumour [131,132]. It may influence MAPK/ERK and PI3K/AKT pathways, which are key factors in inducing MMP-7 expression and human colon cancer HT-29 cell invasion. Lycopene treatment may impact on the increased level and stability of E-cadherin via inhibition of the attachment of a phosphoryl group to AKT, GSK-3β, and ERK 1/2 kinases and inhibition of the nuclear activity of AP-1 and β-catenin. Several studies have also shown the antioxidant activity of lycopene by inducing antioxidant enzymes and phase II detoxifying enzymes.

Bhuvanewari et al. [133] have also observed that lycopene administered in vitro (2.5 mg/kg) can significantly inhibit the hamster buccal pouch carcinogenesis process. This is believed to take place by the multimodal reduction in lipid peroxidation, elevation of antioxidant levels, and enhancement of GSH-dependent enzyme functions, i.e., glutathione reductase, glutathione peroxidase, and glutathione-s-transferase [41,133,134]. Other research confirms that lycopene is the key molecule that increases the activity of antioxidants and antioxidant-responsive elements (EpRE/ARE). Moreover, lycopene may inhibit nuclear factor kappa B activation and adhesion phenomenon through the activity of nuclear factor Nrf2 in endothelial cells, activating of phase II detoxifying-antioxidant molecules. These, in turn, may protect the cells against ROS and electrophilic agents [135,136].

Importantly, another metabolic product of lycopene, apo-8′-lycopenal, may affect tumour spread by inhibiting the level and activity of matrix metalloproteinases, increasing tissue inhibitors of metalloproteinase function, and blocking MAPKs and PI3K-AKT kinases [137]. Additionally, apo-10′-lycopenoic acid (APO10LA), created as a result of the breakdown of lycopene molecule at its 9′,10′-double bond by carotene-9′,10′-oxygenase, exhibits chemopreventive properties by inhibiting oncogenesis and inflammatory reactions. A study of human cancer cells (THLE-2 and HuH7) found APO10LA treatment to exert its chemopreventive effects by increasing hepatic SIRT1 protein; deacetyling SIRT1 targets, decreasing caspase-1 activation and SIRT1 protein cleavage; lowering the expression of proinflammatory interleukins such as TNFα, IL-6, NF-κB, and p65; and increasing STAT3 activation [138].

In addition, lycopene administration was found to downregulate migration, invasion, and adhesion in SK-Hep-1 hepatic tumour cells induced to metastasis by TGF-β; it also reduced MMPs and ROS levels, indicating that inhibition of NOX4, a molecule with a key role in the creation of ROS, determines the antimetastatic properties of lycopene in vitro. Additionally, lycopene appears to inhibit the epithelial–mesenchymal transition in murine CAL-27 oral cancer xenograft model cells, as demonstrated by changes in cadherin activity [75,139], and to slow the in vitro motility of CAL-27 and SCC-9 oral cancer cells via regulation of the PI3K/AKT/mTOR signalling [75,140].

A number of other dietary CTDs belonging to the xanthophylls, including lutein, zeaxanthin, β-cryptoxanthin, astaxanthin, and fucoxanthin, also have antitumour features and impact on local aggressiveness and the spread of malignant cells. All have been observed to reduce the in vitro motility and invasion of tumours of various origin. The anti-spreading oncogenic effect has been related to decreased level of Snail, Twist, fibronectin, N-cadherin, and MMP-2, reduction in the PI3K/AKT intracellular signals and nuclear factor kappa B activity, and enhanced level of TIMP-2. Their anti-invasive and antimetastatic activities have been attributed to decreasing the action of Wnt-1 and β-catenin, fibronectin, MMP-2, vimentin, and VEGF [75,141,142]. Some potential biological intracellular targets of carotenoids in head and neck cancer (HNC) are shown in Figure 3.

### 3.3. Animal and In Vitro Models of Head and Neck Squamous Cell Cancer

Extensive preclinical research has been performed on the stages of carcinogenesis, and the findings play a key role in understanding the tumorigenesis process in HNC-related organ sites. In the literature from the last twenty years, several key in vivo and in vitro scientific works have studied whether carcinogens and/or carotenoids, and other phytochemicals, modulate the proliferation of CSC stem cells or HNSCC cells [38].

Unfortunately, a mismatch exists between epidemiological data regarding the ability of phytochemicals to initiate or modulate cancer and preclinical data in experimental models. This can be caused by a number of reasons, limiting the obtaining of binding conclusions. Many experimental studies are incapable of replicating the complexity of the host system and that of the administered carcinogens, e.g., tobacco, alcohol, and HPV, as well as the habits and doses of the user. It is also very difficult to include the range of exposure sites in the head and neck region in a study; preclinical studies therefore cannot accurately identify the structural and functional differences in metabolic competences and pathways/biomarkers associated with tumorigenesis process affected by chemopreventive factors [38].

#### 3.3.1. Animal Models of HNSCC

Different animal models have been explored to understand the process of HNSCC development. Many have examined the steps in carcinogenesis and tumour dissemination, but most concern the area of HNSCC initiation and promotion, associated with the most well-known HNSCC inducers identified in epidemiological studies, e.g., tobacco, betel quid, alcohol, and HPV [143,144]. Unfortunately, only a few publications have employed animal models to identify environmental HNSCC-inducing agents, such as dietary and/or beverage-derived carotenoids, and to describe their roles in the development of cancer in HNSCC-related organ sites.

In recent years, researchers have used chemically induced, spontaneously developed, transgenic and co-carcinogenic models of oral cancer derived from hamster buccal pouch to examine the development of HNSCC [53,102,145,146]. The first mentioned cellular systems of HNSCC related to the use of popular carcinogens such as 4-nitroquinoline-1-oxide (4NQO), nicotine-derived nitrosamine ketone (NNK), acrylamide, 7,12-dimethylbenz(a)anthracene (DMBA), bovine serum albumin (BSA), bromophenol blue, hydroxyurea, 2-mercaptoethanol, sodium dodecyl sulphate (SDS) *N*,*N*,*N′*,*N′*-tetramethylene diamine (TEMED), trizol and N-nitrosonornicotine (NNN), cigarette smoke, betel nut/leaf extract, 3-(methylnitrosamino) propionitrile (MNPN), alcohol, and tumour-promoting agents such as 12-*O*-tetradecanoylphorbol-13-acetate (TPA), as they mimic human phases of tumorigenesis in animals [38,102].

A recent study by Kavitha et al. [145] based on a hamster buccal model (HBP) of HNSCC induced by the popular carcinogen DMBA indicated that astaxanthin may have chemopreventive effects and that these may act through the stimulation of cytoprotective antioxidants, phase II detoxification molecules, and DNA repair enzymes, activated by Nrf2 and Keap1 proteins. In a subsequent preclinical study, the authors also found astaxanthin induces the intracellular pathway of programmed cell death in a hamster model of oral cancer to inhibit NF-κB and Wnt/β-catenin activity through an inactivation of MAPKs kinases and PI3K/AKT signalling [53]. Other researchers have also used experimental models induced by carcinogens to study the cellular results of CTD activity. A study by Kowshik et al. [102] performed a quantitative RT-PCR, immunoblotting, and IHC research and found that astaxanthin supplementation reduces crucial effects of JAK/STAT factors activity, especially the attachment of a phosphoryl group to STAT3 and its nuclear translocation; they also showed that astaxanthin inhibited the action of STAT-3 target genes relevant for the cell division process, tumour aggressiveness, and angiogenesis, and reduced malignant cell dissemination by downregulating target biomarkers such as cyclin D1, MMP-2 and -9, and VEGF in the HBP carcinogenesis model. Bhuvaneswari et al. [146] reported a negative association between a diet rich in CTDs and a low risk of HNSCC in hamster buccal pouch carcinogenesis. Lycopene, a carotenoid commonly found in tomatoes, shows a high antioxidant activity and has been observed to inhibit tumorigenesis, as evidenced by clinical trials in cell culture and HBP animal models. Research have indicated that lycopene treatment can selectively inhibit cell division cycle by downregulating biomarkers of tumour promotion and proliferation such as PCNA, AgNOR, EGFR, cyclin D1, p53, and p21 and stimulating programmed malignant cell death without affecting noncancerous cells.

Summing up the research on animal models, it is important to emphasize that such carcinogen-induced well-established cancer models are popular and labour-intensive but have many limitations [38,147]. Importantly, while such studies allow prolonged carcinogen exposure, they have limited implications: they prevent the analysis of particular genes in HNSCC progression, and transgenic and xenograft systems do not present an actual picture of human tumorigenesis. Moreover, the most popular animal model, i.e., induced hamster buccal model (HBP), can only mimic the occurrence of HNSCC in the tongue and buccal mucosa subsites. In addition, the cheek pouch model is absent in humans, and the cheek pouch is an immunologically privileged site; furthermore, hamster tumours are usually exophytic, while human HNSCC tumours can be either exophytic or endophytic [38,147].

#### 3.3.2. In Vitro Models of HNSCC

Immortalized HNSCC cell lines from different regions of head and neck, e.g., the pharynx (FaDu), tongue (SCC-4,9,25 or CAL-27), tonsils (FDC-1), and oral area (CAL-27 and WSU-HN6), can also be used for in vitro study. All are accessible and derived from surgically removed human HNSCC tumour tissues [38]. Other lines can also be obtained from transgenic/xenograft animal models; many have been developed from the popular well-established hamster buccal pouch (HBP) model used to study oral cancer tumorigenesis (HCPC1) [38].

A number of recent studies point to the possible chemopreventive effects of dietary- and/or beverage-derived CTDs on novel HNSCC cell lines. Most preclinical experiments concerning target HNSCC cells and cancer stem cells have been focussed on the modulation and/or inhibition of one or more relevant intracellular carcinogenic processes—i.e., cell signalling, xenobiotic metabolizing enzymes, transcription factors, and signalling kinases—are exhibited and described [38]. For example, Ye et al. [148] investigated the cytotoxic effect of lycopene on HNSCC cells to identify possible processes linked to head and neck carcinogenesis. Treatment of FaDu and Cal27 cells with lycopene at a dose of >10 µM for >24 h decreased the proliferation of tumour cells, which was dependent on the dose and duration of stimulation and was connected with an increase in the apoptotic cell population. Lower invasive capacity induced by lycopene was also observed at a dose of 25 µM. This resulted in a significant inhibition of tumour aggressiveness. Scientific research indicates that lycopene can upregulate pro-apoptotic molecules, resulting in reduced MAPK signalling activity. These findings provide an insight into the anticancer abilities of lycopene in HNC tumour cells.

Additionally, the chemopreventive effects of astaxanthin suppress or block the development of tumour cells via different molecular pathways and mechanisms, inter alia, by the NF-κB, PI3K/AKT, ERK/MAPK, and Wnt/β-catenin pathways [38]. This activity is directly related to a decrease in DNA damage and increase in DNA repair, and it blocks tumour progression by decreasing cell proliferation or enhancing apoptosis; it is also believed to modulate the activity of signal tumorigenic pathways [38,149].

It is also worth mentioning two papers from recent years in which researchers analysed the effect of retinoic acid (all-*trans* retinoic particles) on cancer stem cells (CSC) derived from HNSCC patients and the SQ20B cell line [150,151]. CSCs constitute a sparse population among cancer cells and have the ability to differentiate into various tumour cell types. In HNC, these cells are thought to are thought to control tumour onset and development, as well as the progression of neoplastic changes and tumour dissemination, but also MDR phenomenon and recurrence [152]. A study by Lim et al. [150] examined the effect of 0.2 mg all-trans-retinoic acid in DMSO in CSC derived from BALB/c nude mice xenograft tissues. Noticeable inhibition of HNC stem cells was observed from day 5 to day 21 after treatment, caused by downregulation of Wnt/β-catenin signalling. Moreover, targetable signalling events observed in in vitro models of HNSCC were also connected with downregulation of Oct4, Sox2, nestin, and CD44 expression, as well as inhibited tumour sphere formation [150]. Other studies by Bertrand et al. [151] on the SQ20B head and neck squamous cell carcinoma cell line found that all-trans-retinoic acid stimulation can overcome resistance to photon and carbon ion radiation. Moreover, photon and carbon ion irradiation was found to have radio-immunogenic effects on mechanisms of SP-cells and CD44 downregulation in cancer cell lines of HNSCC tumour entities [151].

Two other in vitro studies based on HNC cells also yielded interesting clinical possibilities regarding the anticancer properties of lycopene [140,153]. Tao et al. [153] examined the dose- and time-dependent response of CAL-27 and WSU-HN6 oral cells to 0.25, 0.5, 1, and 2 µM lycopene in in vitro and in vivo experimental systems of OSCC. Lycopene was found to have a regulatory effect on cell migration, apoptosis, and tumour formation. As a result, the cell division cycle was downregulated depending on the duration of action and the dose of lycopene, with the optimum inhibition efficiencies for OSCC cells being noted. Pretreatment of oral carcinoma with lycopene was also found to induce cell apoptosis and to suppress cell migration and tumour growth. Interestingly, the authors observed that lycopene may protect against carcinogenesis by regulating protein expression of IGF1, IGF-BP1, IGFBP3, JUN, and FOXO1 factors. Similar observations were noted by Wang et al. [140], who investigated the antitumour activity of lycopene on the progression of oral cancer in vitro and in vivo. The authors attribute the identified anticancer effect of lycopene to the inhibition of cell division cycle, motility, aggressiveness, apoptosis, and xenograft tumour development in a dose-dependent manner. They also suggest that lycopene has the ability to inhibit the EMT phenomenon and induce programmed death in cancerous cells by downregulation of the PI3K/AKT/mTOR pathway signalling; this is achieved by higher activity of E-cadherin and Bax and lower expression of N-cadherin, and phosphorylated forms of PI3K, AKT, mTOR, and Bcl-2. Hence, it appears that lycopene may regulate HNC cell growth by inhibiting key pathways and so may be a promising factor for the chemoprevention and personalized therapy of this type of cancer.

### 3.4. Studies on the Chemopreventive Efficacy of Dietary Carotenoids in Clinical Trials on Healthy Individuals and Those with Premalignant Conditions

Cancer chemoprevention is defined as the use of natural or synthetic compounds to reverse, suppress, or prevent the carcinogenesis process and tumour progression. To be useful in humans, a chemopreventive agent must have an acceptable safety profile and be effective at doses low enough to not cause significant toxicity. Being sources of carotenoids, fruit and vegetables have great chemopreventive properties due to their potential to inhibit HNC initiation, growth, and expansion [154]. Recent years have seen a growth in interest in the use of new chemopreventive therapies in subjects at increased risk of HNC and for modifying molecular signals/pathways or methods aimed at supporting the immune system. However, studies of chemoprevention strategies targeting HNC have not yet unequivocally confirmed therapeutic properties for oncological experience.

For many individuals at the greatest risk of cancer, the only opportunity to use new chemopreventive methods is to take part in clinical trials. Therefore, it seems important to accurately identify these cases using validated and proven clinical and molecular markers. The safety, tolerability, and the efficacy of chemopreventive agents can be assessed by identifying selected individual or composite endpoints, taking also into account the pharmacodynamics of these substances as well as clinical, histological, and molecular target changes in research systems [38,155]. Novel agents are explored as potential chemopreventive factors in early phase research, and these can eventually lead to the last clinical phase of the trials with endpoints such as cancer onset, progression, and patient prognosis. A recent review estimated the cancerous transition rate of leukoplakia to be approximately 3.5%, ranging from 0.13% to 34% [156,157].

The section introduces the current evidence regarding HNC chemoprevention using dietary carotenoids, as well as the choice of proper and specific endpoints for early phase research. It also examines novel therapeutic perspectives in HNC, especially oral cancer chemoprevention.

#### 3.4.1. Chemopreventive Efficacy of Dietary Carotenoids on Healthy Individuals

Very few, if any, trials reviewing the effect of dietary CTDs, either alone or as the compounds cooperating with other phytochemicals, on the development of HNSCC have been performed in healthy individuals in the last twenty years [38].

#### 3.4.2. Chemopreventive Efficacy of Dietary Carotenoids on Individuals Having Premalignant Conditions

The use of chemopreventive anticancer protocols is a potential alternative treatment strategy for inter alia*,* leukoplakia. Leukoplakia is manifested as epithelial hyperkeratosis and hyperplasia in 90% of cases and also as epithelial dysplasia in the remaining 10%. These pathological lesions are considered precancerous [158]. The primary preventive measures are currently abstinence from tobacco and alcohol abuse and a higher intake of antioxidant nutrients such as carotenoids. However, the risk of oral squamous cell carcinoma remains elevated even after alcohol and tobacco ablactation and may take up to 20 years to reach baseline levels [159]. Chemoprevention can meaningfully limit the possibility of field cancerization and by doing so may prevent the onset of carcinogenesis and thus malignancy. It also may help reduce the morbidity rates associated with performing surgical procedures and may act in the prevention of the more aggressive tumour phenotype or the local and generalized spread of cancer in OSC patients.

However, although a few chemoprevention trials against OSC development have been investigated in preclinical or clinical research, none of them have shown significant benefit and subsequent final acceptance from clinical practitioners. This is believed to be mainly related to the toxicities or clinical ineffectiveness of the proposed chemopreventive agents and the heterogeneity of the target patient population, the inclusion criteria, patient prognosis measures, and endpoints. Nevertheless, studies in the potential of chemoprevention as a modern method of avoiding the development of a fully aggressive form of cancer are ongoing. The majority of these clinical trials, based on the use of purified carotenoids and/or carotenoids with other phytochemicals—i.e., β-carotene with isotretinoin, β-carotene with vitamin A, β-carotene with retinyl acetate—include high-risk subjects with precancerous oral mucosa, i.e., leukoplakia [26,38,160].

Only a few individual articles from the last ten years have described short-term interventional studies or phase I/II clinical trials suggesting that carotenoids may have a direct relationship on premalignant conditions. Sixteen trials are currently underway on the use of various CTDs—including β-carotene alone, lycopene alone, β-carotene with vitamin C, β-carotene with retinyl acetate—in high-risk subjects with precancerous oral mucosa leukoplakia. Only twelve trials have shown positive results, i.e., a reduction in abundance and/or volume of precancerous or malignant changes and a modification of relevant biomolecular parameters. Four of these studies were clinical trials directly involving carotenoids [38].

A few individual studies on the discussed chemopreventive role of carotenoids can be found in the more recent literature; two trials [26,160] found phytochemical treatment to be unsuccessful in decreasing lesion size or the expression of selected histological marked biomarkers. For instance, Nagano et al. [26] describe a double-blinded multicentre randomized controlled clinical trial (RCT) on Japanese subjects with oral leukoplakia. The participants in the experimental arm received 10 mg day^−1^ of β-carotene plus 500 mg day^−1^ of vitamin C over a period of one year, with the placebo arm receiving 50 mg day^−1^ of vitamin C. The primary endpoint in this study was clinical remission at one year. On the other hand, neoplastic transition during the 5-year follow-up of patients was chosen as the secondary endpoint. The overall clinical effect rates were 17.4% in the experimental arm and 4.3% in the placebo arm (*p* = 0.346). Consequently, the relative risk for β-carotene and vitamin C treatment was 0.77 (95% CI: 0.28–1.89, *p* = 0.58). Despite this, neither β-carotene nor vitamin C were effective in terms of clinical response or in preventing the initiation and growth of OSC. The data from this trial contradicts the hypothesis that supplementation of CTDs has a chemopreventive effect for oral leukoplakia [26]. In addition, the variety in p53 protein levels and Ki-67, an indicator of cell proliferation, was also evaluated in the two groups; p53 expression was downregulated in individuals who had clinical effects of CTDs, while the difference in Ki-67 expression was irrelevant between these populations [160]. In addition, five other studies also examined the incidence of cancer following carotenoid treatment; however, only three provided useable evidence. None of the research confirmed that treatment with CTDs decreased the onset of OSC more than placebo—either supplementation of vitamin A (RR = 0.11, 95% CI: 0.01–2.05; 85 participants, one study) or supplementation of β-carotene (RR = 0.71, 95% CI: 0.24–2.09; 132 participants, two studies). Follow-up was in the range of 2–7 years [27].

Papadimitrakopoulou et al. [161] report a study in which 13-cRA (0.5 mg/kg/day) was administered for one year, followed by 0.25 mg/kg/day for two years, or β-carotene (50 mg/day) plus retinyl palmitate (25,000 U/day) for three years. The findings indicated that three-month 0.5 and 0.25 mg/kg b.w./day isotretinon (13-cRA) treatment showed higher efficiency in treating oral cavity leukoplakia compared with vitamin A or vitamin A combined with β-carotene. Progression-free survival for OSC cases did not differ in any of the three subject populations. Side effects were more common in the isotretinoin group. Furthermore, supplementation of β-carotene with retinyl palmitate or retinyl palmitate alone showed no effect, similar to 13-cRA in oral premalignant lesions.

In contrast with the above, Traub [77] and other researchers [162] have indicated that vitamin A, β-carotene, and lycopene have shown clinical resolution rates higher than 50%. The authors indicate that supplementation with vitamin A was clinically effective in treating leukoplakia, as was β-carotene (30 to 90 mg/day), with a response rate from 15% to 71%, and lycopene supplementation (8 mg twice daily).

In addition, β-carotene appears to demonstrate similar effectiveness as retinol at decreasing the levels of micronuclei: the micronucleus test provides immediate information about genotoxic damage, with micronuclei being formed during chromatid or chromosomal breakage—the rate of formation is tied closely to carcinogenesis in the oral cavity. The test itself is a useful indicator of the neoplastic tendency of epithelial cells. Furthermore, β-carotene appears to have a much higher therapeutic index, and as such, should probably be the treatment of choice for this condition. In the quoted study 160 cases with leukoplakia were randomly selected to use the supplementation of vitamin A (300,000 IU/week for 12 months), β-carotene (360 mg/week for 12 months), or placebo. A complete positive response, i.e., remission of oral leukoplakia, was observed in 10% of participants in the placebo group, and in 52% and 33% of cases treated with retinyl acetate and β-carotene, respectively.

Lycopene, a carotenoid present in tomato, also represents a probable chemoprotective natural factor for oral leukoplakia by acting as protector against DNA damage and preventing the initiation and development of dysplasia by negative regulation of the tumour cell division cycle [163]. The significant therapeutic effects of lycopene have been reported by Singh et al. [164] in a phase I randomized controlled trial of lycopene in subjects with precancerous oral leukoplakia over five months. Individuals were supplemented with lycopene (4 mg day^−1^ in group A or 8 mg day^−1^ in group B). Administration was found to decrease the volume of oral changes and the degree of dysplasia; in addition, group A systemic supplementation was more effective than in group B, with 80% of subjects in the former and only 66.2% in the latter showing complete clinical response to treatment [164]. Interestingly, while both groups demonstrated significantly greater responses compared with the reference population, histological assessment was significantly better in the cases taking 8 mg day^−1^ of lycopene.

Similarly, Aung [165] also found lycopene to be efficacious at preventing oral leukoplakia, with treatment with a dose of 8 mg day^−1^ being more effective than 4 mg day^−1^. This efficacy was attributed to its antioxidant properties. Similarly, Zakrzewska [166] report that lycopene brings about significant histological changes in patients with oral leukoplakia. However, the author reports no significant difference in the clinical response to 8 mg day^−1^ lycopene and 4 mg day^−1^; nevertheless, both groups demonstrated significantly greater clinical responses with reference to the control population (*p* < 0.01). Interestingly, the 8 mg lycopene treatment yielded significantly better histological response than the 4 mg treatment (*p* < 0.05) and the reference group (*p* < 0.001).

Similar conclusions were recorded by Johny et al. [167] in a study of the effectiveness of the use of lycopene and the addition of hyaluronidase to lycopene in the course of clinical oral submucosal fibrosis. Two groups of individuals were distinguished in the study: the first were given 16 mg daily lycopene as two 8 mg doses for three months, while the other received lycopene with hyaluronidase as an intralesional treatment of 1500 IU twice weekly for three months. The rest of the patients were given placebo capsules. As a result, the lycopene and lycopene–hyaluronidase groups demonstrated a statistically significant improvement in oral function and reduction in discomfort compared with the placebo group; however, the lycopene–hyaluronidase treatment scheme did not confirm any statistically positive relationship in comparison with the lycopene supplementation alone.

Research data show that lycopene may be a beneficial chemopreventive agent in the treatment of some precancerous lesions in the oral cavity and may be an effective agent protecting and inhibiting the carcinogenesis of OSC. However, randomized controlled trials with larger target groups are prerequisites for the administration of lycopene in prophylaxis and routine treatment protocols for these lesions [168].

It is also worth mentioning one more aspect of the use of chemopreventive carotenoids. Several attempts have been made to assess their relationship with the later, more aggressive development of neoplastic cell tumours and to determine the role of carotenoids in these processes; for instance, studies have examined their role in the risk of developing a second primary tumour (SPT), the most common assessment endpoint. However, it should be emphasized that only a few observational and interventional studies have attempted to assess the chemopreventive efficacy of CTDs in HNSCC [38]. Carotenoids have not been found to reduce the incidence of secondary primary tumours in HNSCC cases, at various doses or either alone or in combination, i.e., β-carotene alone, β-carotene plus isotretinoin (13-cis-retinoic acid), and retinyl palmitate plus β-carotene [38]. Importantly, few publications have attempted to use different doses of carotenoids for chemoprevention of locoregional recurrences and distant metastases.

To summarize, it is important to emphasize that it is not possible to identify a specific carotenoid or other phytochemical with a direct, precise potential link to the chemoprotection in HNC.

### 3.5. Studies on the Role of Dietary Carotenoids as Predictors of Cancer Risk, Progression, and Prognosis in Patients with Diagnosed Head and Neck Cancer

The term diet refers to the daily consumption of food and drink by an organism. The diet is a source of the nutrients necessary for normal intracellular biochemical reactions and for a variety of key signalling pathways, and diet is recognised as playing a critical role in various metabolic diseases, cardiovascular diseases, and osteoporosis, among others [21]. The significance of proper nutrition on tumour initiation and development, however, remains unclear in cancers of various origin; this is due to the numerous environmental and genetic exposure factors, different carcinogenesis sites in the head and neck region, and the difficult to estimate duration of exposure to carcinogens. Although the etiological association between both diet and nutrition and the onset and growth of cancer is still unknown, the results from recent years unequivocally indicate a significant relationship between diet and both carcinogenesis and subsequent stages of cancer development. The Third Expert Report of the World Cancer Research Fund (WCRF) and the American Institute for Cancer Research (AICR) has confirmed that diet and nutrition are significant parameters in the carcinogenesis of a range of human cancers, including HNC and stomach, kidney, lung, breast, liver, and prostate cancer [11,22]. Moreover, recent works also emphasize the risks to HNSCC subsites associated with the physical contact of nutrients and compounds with the upper respiratory epithelium; of these, the most convincing relationship was found with the OSC [11,169].

Plants contain a range of bioactive components, including carotenoids [22]. Most of these dietary compounds act as factors showing proven antioxidant and anti-inflammatory properties and an immune-regulating effect and can directly influence tumorigenesis. The antioxidant activity of carotenoids may reduce the presence of ROS, inhibit oxidation-related changes, favour DNA stabilisation, and stimulate the transcription of targeted genes. Bioactive food ingredients are known to influence a range of proinflammatory pathways, growth factor translation and transcription process, the cell division cycle, mRNAs expression, and epigenetic changes in crucial molecules e.g., NF-κB, Wnt, Akt, MAPK, and Notch pathways [23,169,170].

This section presents some key works regarding the topic of dietary intake and nutrition and some of the key research on the role of dietary CTDs as predictive factors in HNC. For easier presentation, the publications are divided into analyses of carotenoid intake and pretreatment–posttreatment serum carotenoid concentration as predictors of HNS risk and as predictors of survival and cancer recurrence.

#### 3.5.1. Carotenoid Intake and Head and Neck Cancer

A wide range of studies have confirmed an inverse relationship between the intake of CTDs present in fruits and vegetables and the development of HNC [16,17,169,171,172]. The most extensively analysed carotenoids include β-carotene, lutein, lycopene, and zeaxanthin. However, some studies also indicate that carotenoids have a negative or insignificant effect on carcinogenesis in HNC [19,173,174].

It is also worth emphasizing that a range of professional validated food–frequency questionnaires (FFQ) (e.g., the semi-quantitative Harvard FFQ) and/or index-based dietary patterns (e.g., the Healthy Eating Index-2005, HEI-2005; the Healthy Eating Index 2015, HEI-2015; and alternate Mediterranean Diet Score, aMED) or the National Health and Nutrition Examination Survey (NHANES) results are used in the studies. These tools allow the most objective analysis of the role of diet and nutrition compounds, including carotenoids, on the risk, onset, and progression of cancer and the estimation of circulating nutrient levels [175,176,177].

When discussing the impact of CTDs on the risk, progression, and prognosis in HNC, it is impossible not to acknowledge the very important multicentre collaborative INHANCE project. The International Head and Neck Cancer Epidemiology venture started its activity in 2004 to elucidate the aetiologic agents of HNC through extensive analyses of individual and multipopulation outcomes from large-scale molecular epidemiology case–control studies. Over the past 18 years, INHANCE has accumulated over 35 studies (around 25,000 cases and 37,000 controls) covering populations from Europe, the United States, South America, Australia, and Asia. More than 20 nutrients have been analysed in multicentre epidemiological studies to reveal a posteriori models of dietary behaviour and their relationship to the onset of HNC carcinogenesis. Most importantly, the large sample size described in INHANCE research groups provides a clear evaluation of the impact of dietary components on the development of HNS in never-smoked individuals to avoid smoking-related effects [14,15,178,179,180,181,182,183,184]. Most research from the INHANCE consortium collected data on nutrition using a study-specific FFQ; this was used to determine daily dietary habits and to estimate the reference period that preceded the diagnosis of malignant neoplasm in HNC patients.

A key study by Leoncini et al. as part of INHANCE [182] indicates the presence of a relationship between HNC development and the consumption of CTDs, i.e., β-cryptoxanthin, lycopene, and lutein with zeaxanthin, as well as carotenoids in combination, including β-carotene equivalents and total carotenoids. This epidemiological analysis consisted of 10 case–control studies and encompassed 18,207 subjects (4414 with oral and pharyngeal cancer, 1545 with laryngeal cancer, and 12,248 controls), categorized by quintiles of CTD consumption from natural sources and expressed as the index: μd/day. The population was originally from Europe, Japan, and North America in which data on CTD intake were available from study-specific questionnaires. Comparing the highest with the lowest quintile, the risk reduction related to total CTD consumption was 39% (95% CI: 29–47%) for oral/pharyngeal cancer and 39% (95% CI: 24–50%) for laryngeal cancer. The consumption of CTDs, i.e., β-carotene, β-cryptoxanthin, lycopene, and lutein with zeaxanthin, was associated with an almost 18% reduction in the risk of oropharyngeal carcinogenesis (95% CI: 6–29%) and a 17% reduction in the incidence of laryngeal cancer (95% CI: 0–32%). The protective effect of CTDs on HNC risk was higher for β-carotene and lower for lycopene and differed by geographical region, being stronger in Europe. Importantly, an interesting relationship was found between total carotenoid intake, defined as the total amount of α-carotene, β-carotene equivalents, β-cryptoxanthin, lycopene, and lutein with zeaxanthin, and the effect of alcohol and tobacco use. The odds ratio for the combination of low CTD consumption and alcohol or smoking abuse vs. high CTD consumption and low alcohol or tobacco use ranged from 7 (95% CI: 5–9%) to 33 (95% CI: 23–49%). A low intake of CTDs was related to a higher risk of carcinogenesis in the head and neck region, including the oral cavity, pharynx, and larynx in heavy drinkers (≥5 drinks per day) and with a higher risk of oropharyngeal (but not laryngeal) cancer in current smokers (>20 cigarettes/day).

Leoncini et al. [183] also describe a systematic review and meta-analysis of epidemiological INHANCE data to explore whether the consumption of specific CTDs derived from food, as well as combined CTDs, is associated with the onset of HNC according to cancer subsites. Investigators analysed 16 scientific publications, of which 15 were case–control studies and 1 was a prospective cohort study. The reduced risk of cancer development associated with β-carotene equivalents intake was 46% (95% CI: 20–63%) for OSC and 57% (95% CI: 23–76%) for laryngeal malignancy. Lycopene and β-cryptoxanthin also decreased the risk of carcinogenesis in the laryngeal epithelium; ORs for the highest category compared with the lowest CTD consumption were 50% (95% CI: 11–72%) and 59% (95% CI: 49–67%), respectively. Lycopene, α-carotene, and β-cryptoxanthin were significantly associated with a 26% reduction in the incidence of oropharyngeal carcinogenesis (95% CI: 2–44%).

Another key study is that of Chuang et al. [15], which examined the association between diet and HNC risk using INHANCE data. This study analysed 22 case–control studies with 14,520 patients and 22,737 controls, most of whom were drawn from American and European pooled data. An inverse relationship was confirmed between HNC onset and higher frequency consumption of fruit (4th vs. 1st quartile OR = 0.52, 95% CI: 0.43–0.62, *p*_trend_ < 0.01) and vegetables (OR = 0.66, 95% CI: 0.49–0.90, *p*_trend_ = 0.01), especially with the highest intake of fresh or cooked carrots, fresh tomatoes, and tomato sauces being the source of carotenoids (the overall *p* for the heterogeneous consumption of these vegetables was *p* < 0.01). In summary, the authors indicated that increased intake of green salads, lettuce, and fresh tomatoes (>seven times/week) was associated with a lower carcinogenesis of HNC.

Similar results were also obtained from populations in Taiwan and Brazil, in whom a decreased risk of HNC development was connected with higher fruit and vegetable intake [16,17]. Both studies showed that diet is a relevant factor in the genesis of HNC due to the protective properties of bioactive compounds of natural food. De Podestá et al. [17] report a multicentre case–control study, a large cohort study of the International Consortium on Head and Neck Cancer and Genetic Epidemiology Study coordinated by the IARC, performed on a final sample of 1740 subjects (847 cases and 893 controls) including malignant neoplasms of the head and neck of various origins. The authors noted that the daily intake of fresh tomatoes, as a lycopene source, significantly decreased the risk of OSC by 72% (OR 0.28, 95% CI: 0.14–0.56, *p* < 0.001). Likewise, carrot consumption, as a good nutritional source of β-carotene, reduced the odds ratio of oropharyngeal carcinogenesis by 86% (OR 0.14, 95% CI: 0.04–0.44, *p* = 0.001). The authors indicated that lycopene and β-carotene, present in tomatoes and carrots, respectively, demonstrate antioxidant activity, hindering the development of human tumour cells and inhibiting the secretion of proinflammatory IL-8 induced by tobacco carcinogens; this may explain the possible relationship found in this study.

Similar observations were reported by Chang et al. [16]. This case–control study investigated the relationship between dietary habits and HNC incidence using data from 838 HNC cases and 998 controls. HNC risk, adjusted for sex, age, education, and use of alcohol, betel quid and cigarettes, was found to be related to a wide range of compounds contained in food. In particular, the authors noted that persons who did not use fresh fruits and vegetables in the daily diet had twice the risk of HNC carcinogenesis compared with those who did (OR = 2.24, 95% CI: 1.54–3.25). Interestingly, those who only consumed fresh vegetables every day had a 1.45 higher risk of HNC compared with individuals with a daily consumption of both fresh vegetables and fruits (OR = 1.45, 95% CI: 1.14–1.84), regardless of the above-mentioned adjusted parameters. These findings further prove an inverse relationship between HNC risk and fresh vegetables/fruits intake as rich sources of CTDs.

In accordance with a study published by De Vito et al. [180] based on a meta-analysis of individual-level results from seven case–control studies (3844 patients; 6824 controls), providing information on 23 dietary compounds yielded both shared and study-specific a posteriori patterns. The researchers confirmed that the levels of antioxidants such as carotene were inversely related to oropharyngeal carcinogenesis (OR = 0.57, CI: 0.41–0.78, *p* for trend = 0.003, highest vs. lowest score quintile). Additionally, a linear tendency for laryngeal cancer initiation was clear for the antioxidant vitamins deficiency, for which protective effects were observed from the second quintile category onward, although the CI for the last quintile category included 1 (OR = 0.62, 95% CI: 0.37–1).

A posteriori patterns were also identified by Edefonti et al. [184] by standard principal component factor analysis of five case–control revisions (2452 patients and 5013 controls) participating in INHANCE; the studies provided key data on common nutrients, and the results were standardized and analysed as a single dataset. The cases were characterised by an aggressive tumour phenotype of the oral cavity or of the oropharyngeal, hypopharyngeal, pharyngeal, or laryngeal region. Subjects with malignant tumours of the salivary glands or of the nasal cavity/ear/paranasal sinuses were excluded. However, the conducted analysis allowed the indication of a slightly shorter list of nutrients, but it also obtained relevant information on both single CTDs and their combinations (i.e., total carotene or β-carotene equivalents and/or single carotenoids) which were not completely comparable with those given by De Vito [180]. Edefonti et al. [184] identified three major dietary patterns, including antioxidant dietary components such as vitamins and fibre, whose occurrence was inversely associated with the risk of oropharyngeal malignant tumours (OR = 0.57, 95% CI: 0.43–0.76 for the highest vs. the lowest score quintile).

A few other studies have also summarized data on β-cryptoxanthin and lycopene in relation to head and neck carcinogenesis [185,186]. Bidoli et al. [185] found the highest category of β-cryptoxanthin intake to have a protective effect against oropharyngeal malignancies (OR = 0.46; 95% CI: 0.29–0.74; I^2^ = 51.8%; *p* = 0.015) and for laryngeal carcinogenesis (OR = 0.41; 95% CI: 0.33–0.51; I^2^ = 0.0%; *p* = 0.049) compared with the lowest intake. Similarly, the meta-analysis confirmed that lycopene has protective effects against oropharyngeal cancer (OR = 0.74; 95% CI: 0.56–0.98; I^2^ = 14.5%; *p* = 0.032), as well as against laryngeal cancer (OR = 0.50; 95% CI: 0.28–0.89; I^2^ = 65.9%; *p* < 0.05). In addition, a multicentric case–control study performed by Bravi et al. [186] in Italy and Switzerland determined the importance of nutrient supplementation in malignant lesions of the oral cavity and throat in a group of 768 patients with histologically confirmed SCC and 2078 hospital controls. Significant inverse relations were noted for α-carotene (OR = 0.51, for the highest vs. the lowest quintile of intake), β-carotene (OR = 0.28), β-cryptoxanthin (OR = 0.37), and lutein and zeaxanthin (OR = 0.34) with regard to the incidence of cancer.

A large study by the same group under the leadership of Bravi [14] summarized results from a recent review of 17 INHANCE-based publications published from 2015 to 2020. The identified studies examined the most common and important risk factors associated with HNC; in particular, it was found that a diet related to fruit and vegetables appeared to have a positive impact on the incidence of this type of malignant neoplasm. The INHANCE group published risk prediction models and summarized follow-up results on their studies. Researchers used study-specific FFQs to determine subjects’ habitual dietary habits in the reference period prior to cancer clinical onset, while the controls received an interview. Whenever possible, information on dietary compounds was obtained by using the psychometrically validated food–frequency questionnaires with data from country-specific food composition databases (i.e., Gnagnarella, Salvini, and Parpinel 2015; US Department of Agriculture 2013) [14,178,179]. The summary of the latest INHANCE analysis indicated that a daily healthy diet based on foods rich in antioxidant compounds such as fruits and vegetables, has an inverse impact on laryngeal carcinogenesis; such a dietary pattern is a rich source of antioxidant dietary CTDs [14].

The observations obtained via INHANCE publications have supported the development of head and neck cancer prevention strategies based on balanced consumption of foods rich in dietary carotenoids; they have also demonstrated the interactive effects of dietary compounds on their bioavailability and the influence of circulating levels of nutrients on HNC disease risk [14,15,178,179,180,181,182,183,184]. These interesting results clearly show that such international, multicentre collaboration allows for measurable results and conclusions and can contribute to increasing knowledge about dietary intake and nutrients in HNC; in addition, such efforts can contribute to the development of chemopreventive therapeutic strategies. The INHANCE consortium results may also inform future national dietary recommendations in various countries including Europe and the US, and the results may serve as a point of reference for future research.

Similar observations were observed in an Italian study of the relationship between macro- and micronutrients and the incidence of another type of HNC, nasopharyngeal carcinoma, in a southern European population [187]. This hospital-based case–control study was conducted on 198 NPC patients and 594 cancer-free controls. All participants were Caucasian. NPC risk was confirmed to be inversely associated with CTD consumption, especially carotenes (OR for highest vs. lowest quartile = 0.46; 95% CI: 0.26–0.79; *p*_trend_ < 0.01), α-carotene (OR = 0.57; 95% CI: 0.33–0.97), and β-carotene (OR = 0.42; 95% CI: 0.24–0.75). The findings directly indicate that CTDs have a protective impact on incidence of nasopharyngeal carcinoma in a low-risk cohort, undoubtedly pointing to the positive importance of a healthy diet rich in fruits and vegetables in HNCs of various origin. Such relationships between the development of NPCs and dietary habits were also observed in the high-risk group and in the US [22,188]. Kasum et al. [188] also report an inverse relationship between the incidence of this type of cancer and the consumption of CTD-rich “orange, yellow, and red” vegetables. The representative scientific publications confirming an inverse association between carotenoid intake and HNC risk are shown in Figure 4.

Unfortunately, other case–control studies have not confirmed any substantial relationship between the amount of CTDs consumed (β-carotene equivalents) and HNC risk. For instance, in a case–control research study performed in Japan, no significant inverse relationship was noted between β-carotene equivalent intake and cancer of the pharynx (OR = 0.61; 95% CI: 0.31–1.20); a nonsignificant inverse relationship was also confirmed when oral and pharyngeal cancer were considered together [189]. Three subsequent studies did not confirm the relationship between the consumption of α-carotene and the presence of HNC of various origins [185,190,191]; however, a strong negative relationship was noted between high α-carotene supplementation and the occurrence of cancer of the oropharyngeal region (OR = 0.57; 95% CI: 0.41–0.79; I^2^ = 0.0%; *p* = 0.35), and no significant relationship was noted with laryngeal malignancy (OR = 0.46; 95% CI: 0.20–1.06; I^2^ = 83.1%). Furthermore, another case–control study reported no significant relationship between lutein and zeaxanthin consumption and oropharyngeal carcinogenesis (OR = 0.51; 95% CI: 0.22–1.18; I^2^ = 83.0%) and laryngeal cancer incidence (OR = 0.60; 95% CI: 0.27–1.32; I^2^ = 0.0%), as well. Similarly, another case–control study by US researchers did not find a significant association with HNC when taking lutein supplementation into account (OR = 0.95; 95% CI: 0.52–1.73) [185].

In addition, a further large cohort analysis displayed no significant inverse relationship between β-carotene intake and the incidence of oropharyngeal carcinoma (OR = 0.57; 95% CI: 0.14–2.38, I^2^ = 93.9%) [190]. Furthermore, de Munter et al. [191] investigated the relationship between the common CTD intake (α-carotene, β-carotene, lutein with zeaxanthin, lycopene, and β-cryptoxanthin) and the risk of HNC in general and malignant subtypes in the Netherlands Cohort Study; the large cohort comprised 5000 participants (2411 men and 2589 women) randomly chosen from the total population of individuals at baseline to design a subcohort in which the risk of carcinogenesis was estimated (based on the individual characteristics of the study participants and the duration of observation—person-years). The average degree of daily intake of α-carotene, β-carotene, lycopene, and β-cryptoxanthin, and lutein with zeaxanthin was estimated based on the results of a 150-item FFQ. The researchers noted that in adjusted analysis taking into account age and sex, *p*_trend_ was only substantial for the relationship of β-cryptoxanthin consumption and HNC in general (*p*_trend_ = 0.001) and carcinoma of the larynx (*p*_trend_ = 0.002). Nevertheless, no associations were confirmed between supplementation of α-carotene, β-carotene, lycopene, and lutein with zeaxanthin and the overall incidence of head and neck cancers and its most-common types, i.e., oropharyngeal, laryngeal, and oral cavity cancer.

In addition, the Alpha-Tocopherol, Beta-Carotene Cancer Prevention (ATBC) study, a double-blind, placebo-controlled trial performed in southwestern Finland did not confirm any association between dietary CTDs consumption and the incidence of HNS [19]. The researchers investigated whether daily intake of 50 mg dl α-tocopherol acetate and/or 20 mg β-carotene decreased the risk of oral, pharyngeal, oesophageal, and laryngeal cancers, as well as mortality from the conditions. A total of 29,133 male smokers who did not have any HNCs at baseline were randomized into a supplementation regimen for five to eight years. The researchers found neither agent to have any effect on the risk of upper aerodigestive tract carcinogenesis. In laryngeal cancer cases, however, the conducted analyses clearly indicated a correlation between β-carotene supplementation and a protective effect on the risk of early-stage carcinomas (stage I) and the relative risk (OR = 0.28; 95% CI: 0.10–0.75). As such, the obtained data do not indicate the chemopreventive action of β-carotene intake on malignancies in this region, although β-carotene supplementation may affect the frequency of certain subtypes of laryngeal tumours.

The differences in these findings could be accounted for by variation in study design: recall bias, selection bias, and reversed causation are more likely to occur in case–control designs or because of hospital-based controls, who may not be representative of the general population. Moreover, the researchers note clear heterogeneity in the incidence of HNC when considering different levels of exposure to tobacco smoking and alcohol drinking. It is possible that smoking status and alcohol abuse may modify the relationship between carotenoid consumption and head and neck cancer risk. However, as the numbers of individuals in the strata was relatively low, suggestion about the possible interaction between tumour risk and CTD intake remain unclear. Additionally, the fact that differences exist between HNC subtypes is also important. These could be explained by variation in exposure due to the location of the tumour: the oropharyngeal region is most exposed to the chemopreventive effects of CTDs, which could yield greater expected effects than in the case of laryngeal cancer. Lastly, the studies incorporated various study designs, including various degrees of exposure to oncogenic factors used to define the estimated categories.

Epidemiological and clinical studies in recent years have provided strong evidence that that various chemopreventive agents included in the diet, and hence possibly dietary carotenoids, have pro-tumour features and have the potential to facilitate invasion and metastasis in human cancers [11,21]. Although some ambiguity remains regarding the clinical importance of diet in HNC progression and metastases, the Third Expert Report of the WCRF and the AICR has lately indicated strong positive correlation between diet and nutrition and the risk and aggressiveness of HNC, as well as their role in the onset and progression of stomach, lung, liver, kidney, breast, and prostate cancer [22]. Such studies have been performed in various countries [16,17,171]. Detailed analyses have shown a negative correlation between higher fruit and vegetable intake and the incidence of HNSCC. In particular, subjects consuming greater amounts of CTDs in the diet had a 39% lower risk of HNC carcinogenesis than those with low carotenoids intake [182]. Moreover, studies have indicated that the properties of phytochemicals can influence numerous cell signalling pathways linked to proinvasive and prometastatic activities at multiple stages of HNC progression, e.g., nuclear factor kappa B, AKT, Notch, MAPKs, and Wnt pathways [23].

When discussing the role of dietary carotenoid intake on the risk, progression, and prognosis of HNC, it is also necessary to note the very important role played by tobacco smoking and alcohol drinking on the final effect of dietary score. For instance, Chen and co-authors [192] introduced a simple nutrition index to accurately assess the role of dietary chemopreventive agents in the incidence of OSC in a large case–control study of 930 cases and 2667 controls in the southeast of China; the resulting dietary score was significantly related to the onset and progression of OSC based on Chinese dietary patterns. In addition, the researchers observed significantly stronger interactions between the nutrition index and tobacco or alcohol abuse for carcinoma of the oral cavity; the OR_multiplicative_ values were 1.67 (95% CI: 1.54–1.82, *p* < 0.001) and 1.49 (95% CI: 1.38–1.62, *p* < 0.001), respectively. The study also presented a substantial relationship between dietary score and the incidence of oral cancer, when taking into account the modifying effect of tobacco and alcohol consumption with the amplifying effect. The authors suggest that the activity of carcinogens form tobacco and the co-carcinogenic effects of alcohol may disrupt the metabolic pathways of some CTDs, especially α-carotene and β-carotene, resulting in a reduction in the serum concentration of micronutrients in individuals.

Similarly, correlations between vegetable consumption and an incidence of HNC were observed mostly in smokers or heavy drinkers but not in never-smokers and light drinkers [15]. An explanation for these clinical observations in never-smokers could be that nutrients in plant diet, such as vitamins, flavonoids, and also CTDs could affect the carcinogenesis in smokers due to their antioxidant, anti-inflammatory, and immune regulatory functions. Hence, the effects of reducing smoke-induced oxidative damage or inflammatory responses by these nutrients can only be characteristic of smokers. An alternative interpretation is that the aetiology of HNSCC in never-smokers may be associated with the influence of other oncogenic factors, i.e., human papilloma virus infection, an established risk agent for OSC; this may have an impact on the relationship between dietary CTD consumption and the incidence of HNC. It has been found that in viral carcinogenic cancers, the human papillomavirus-16 (HPV16) may directly affect the relationship between micronutrient consumption and HNSCC [193,194,195,196,197].

Elsewhere, it was found that greater consumption of fruits and vegetables were linked to a lower incidence of HNC among smokers, regardless of age; in addition, the obtained data highlight a significant and crucial effect of the combination of decreased fruit and vegetable intake with exposure to tobacco and alcohol carcinogens, with 10- to over 20-fold greater risks of HNC [198]. Similar observations have been noted by the Head and Neck 5000 prospective clinical cohort study (HN5000) [199], based on data from 2202 participants; the study took into account the disturbances resulting from the differences in the intensity of environmental exposure within oral, oropharyngeal, and laryngeal cancer subgroups. It was found that 30% and 40% of cohort members consumed a diet high in fruit and vegetables as a rich source of carotenoids. In the other adjusted model for head and neck regions, high fruit and vegetable consumption was related to a 44% decrease in mortality rates (HR = 0.56, CI: 0.44–0.73, *p*_trend_ < 0.001). In addition, high vegetable consumption in HNC individuals was associated with an increase in overall survival (HR = 0.79, CI: 0.61–1.03, *p*_trend_ = 0.04); in particular, a 54% improvement in overall survival was noted for laryngeal cancer. However, these relationships were not confirmed following adjustment for health risk behaviours (HR = 0.91, CI: 0.67–1.23, *p*_trend_ = 0.55) for fruit intake. Moreover, no dose–response relationship was seen between vegetable consumption and overall survival in cancer of the larynx. There was no evidence of any interaction between smoking and vegetable consumption (*p* values ranged between 0.33 and 0.94) or between alcohol abuse and vegetable intake (*p* values ranged between 0.13 and 0.96). The representative scientific articles confirming a positive relationship, or none, between carotenoid intake and HNC risk are shown in Figure 5.

Importantly, Leoncini et al. [182] report interesting observations between total carotenoid intake—defined as the unweighted sum of α-carotene, β-carotene, lycopene, β-cryptoxanthin, and lutein plus zeaxanthin—and the effect of alcohol and tobacco use. The ORs for the cumulative effect of low CTD consumption and alcohol or tobacco abuse vs. high CTD supplementation and low alcohol or tobacco consumption ranged from 7 (95% CI: 5–9) to 33 (95% CI: 23–49). A low carotenoid intake was related to a higher incidence of oropharyngeal and laryngeal cancers in heavy drinkers (≥5 drinks/day) and with a higher risk of oropharyngeal (but not laryngeal) cancers in current smokers (>20 cigarettes/day).

Although several potential mechanisms of CTDs chemopreventive activity can be identified, it is commonly believed that these phytochemicals can act as provitamin A, facilitating cellular differentiation and proliferation. Furthermore, the bioavailability and absorption of the synthetic form of CTDs are also different from natural sources. Interestingly, several studies have confirmed a higher incidence of neoplasms in people taking β-carotene preventively [200]. For example, The Beta-Carotene and Retinol Efficacy Trial (CARET) study, as well as others, noted that the supplementation of β-carotene with vitamin A was linked to increased incidence of malignant lesions among current smokers and heavy drinkers [201]. Furthermore, the researchers also noted that β-carotene could initiate an alternative mechanism with a pro-oxygenic effect; this may be linked to its strong interaction with unhealthy lifestyle behaviours in the population of heavy smokers and heavy drinkers.

It is also worth noting that the strength of these INHANCE consortia multicentre analyses from the last twenty years is that the findings indicate that individual and combined carotenoids from diet appear to have an influence on the regions and subsites of head and neck cancers. Nevertheless, the studies do have limitations. First, although most researchers have modified their observations by considering previously identified risk factors for HNSCC, it is impossible to rule out the factors not related to diet and nutrition, e.g., HPV infection. Second, although tobacco smoking and alcohol abuse are crucial risk factors for HNC, the authors of meta-analyses were unable to classify individuals according to smoking and drinking status, which may also affect the obtained observations and final conclusions. Moreover, the case–control study format of the publications also has limitations: although case–control researchers have an advantage of being able to start from a selected endpoint, they are however unable to determine the sequence of events, which reduces the possibility of identifying causality. Long-term prospective observations appear to have greater predictive value. They have several time points and provide more reliable information on the association between the exposure to chosen carcinogenic agents and the course of the malignancy. Nevertheless, among scientific works from the last twenty years, few studies on HNC appear to have been performed.

Another relevant aspect of patients with HNC is the fact that despite advances in the management of the disease, treatment is commonly associated with toxicities that compromise dietary intake and nutritional status [202,203]. The HNC patients have to face challenges specific to this group of individuals, as compared with other malignancies, affecting their functional state (swallowing disorders, xerostomia, etc.) and determining their quality of life. It is obvious that this situation must have an impact on dietary intake and malnutrition at baseline due to an underlying dysphagia. As a result of the use of intensive local and systemic therapy in people with advanced neoplastic tumours of the head and neck region, this group of patients experiences many ailments that affect the ability to eat sufficiently large portions of food with an appropriate nutritional composition; this is often the cause of the so-called *nutrition impact symptoms* (NISs) [204]. Common symptoms of NISs include dysphagia, xerostomia, and chewing and swallowing difficulties, which often lead to restricted food consumption, malnutrition, weight loss, difficulties in healing, and increased susceptibility to infections [204,205,206,207]. The researchers have conducted interesting research on lycopene supplementation as a way to reduce the risk of inflammatory local processes of mucosa in HNSCC subjects who undergo intensive radiation therapy. The increase in lycopene consumption was significantly related to the protective effect of this CTD against radiation damage in experiments carried out in animal models; lycopene supplementation after the use of external beam radiation therapy reduced the frequency of weight loss and the parameters of oxidative stress [208]. Nutrition-based multicentre interventions to reduce the incidence of NIS in people with HNSCC have also been postulated; the use of such clinical interventions may contribute to improvement in the quality of life, functional status, and implementation of healthy dietary habits in this population [209,210].

Another aspect of research into the importance of carotenoid intake in HNC risk should also be highlighted. Interestingly, some clinical studies using mineral supplements have not shown a strong relationship between isolated nutrients and the incidence of HNC; this may indicate that a single, isolated food ingredient will not produce the synergistic effect of the numerous chemopreventive compounds present in whole, natural, unprocessed foods. The use of concentrates containing the right proportion of fruit and vegetables in patients with HNC may be an alternative and sufficient intervention to increase the intake of CTDs, vitamins, minerals, and phytochemicals [18,19]. Chainani-Wu reports that vegetables contain high levels of micronutrients with anticancer properties, such as carotenes, lycopene, and other antioxidant vitamins; in some cases, the combination of several nutritious compounds increases their the clinical chemopreventive effect [174]. A randomized, double-blind, placebo-controlled trial by Datta et al. [173] evaluating the clinical effects of Juice PLUS+ (JP), a commercial product with multiple FV concentrates, found that after 12 weeks, patients on JP had markedly increased serum α-carotene (*p* = 0.009), β-carotene (*p* < 0.0001), and lutein content (*p* = 0.003). Therefore, future perspectives for the HNC population should apply to the use of long-term interventions, such as CTD, vitamin, and mineral supplementation to demonstrate the chemopreventive properties of these healthy nutrients.

Similarly, Li et al. [176] prospectively evaluated the relationship between two index-based dietary patterns, i.e., the HEI-2005 and the aMED, and the incidence of HNSCC. It was observed that the results of the HEI-2005 and aMED indexes were negatively associated with HNC. Similarly, Filomeno et al. [211] report the Mediterranean diet to be inversely associated with a risk of oral and pharyngeal cancer for increasing levels of MDS: the ORs for subjects with six or more MDS compounds was 0.20 (95% CI: 0.14–0.28, *p*_trend_ < 0.0001) compared with those with two or less.

Elsewhere, the diet quality of 42 HNC survivors, at least six months postradiation, was evaluated in a cross-sectional study by Crowder et al. [177]. The score was calculated using the HEI-2015 index and compared with NHANES data. HNC survivors reported a significantly lower total HEI-2015 score compared with the healthy NHANES controls; the recorded observations clearly indicate that chronic malnutrition and the lack of an adequate level of nutrients in the body, according to the previously described NISs syndrome, activate patients to a greater desire and ability to replenish dietary phytochemicals, including CTDs. Additionally, in a study of 600 nasopharyngeal cancer patients and 600 matched controls, Wang and collaborators [212] report that after adjustment for various lifestyle and dietary factors, higher diet quality indexes on the HEI-2005, the aHEI, and the DQI-I, but not on the aMed score, were significantly associated with a lower incidence of this type of cancer (p_trends_ < 0.001).

Many studies using various questionnaires have underlined the need for reliable estimation of dietary patterns in reducing the risk of development and progression of HNC [176]. Although the role of consumed CTDs in head and neck cancers has relatively rarely been discussed and the results are ambiguous, several prospective studies have still found fruit and vegetable intake to have inverse associations with HNC risk, particularly for nutritional substances that are rich in CTDs [22,176,213]. The above-mentioned questionnaires may be helpful tools in the assessment of nutritional deficiencies in patients with HNC. Two examples could be HEI-2005 and aMED indexes, which were related inversely to the incidence of HNC; these may indirectly indicate that the dietary intake and the serum levels of carotenoids may play a significant role among HNC patients.

#### 3.5.2. Serum Pretreatment and Posttreatment Carotenoid Concentrations in Head and Neck Cancer Risk and as Predictors of Survival and Recurrence

The researchers from the European Prospective Investigation into Cancer and Nutrition (EPIC) consortium have introduced evidence that consumption of specific nutrients, as measured by various questionnaires, are proven predictive factors of the blood levels of some chemopreventive agents, particularly CTDs [214]. The CTD concentration in serum is therefore a promising indicator of nutrition consumption, with β-carotene, lycopene, lutein, and zeaxanthin being the most extensively researched carotenoids. Interesting research and observations by Al-Delaimy et al. [214] clearly showed that healthy subjects who followed a low-carotenoid diet (<0.4 mg/d) had plasma CTD levels <60% of the initial levels observed at the beginning of the experiment. After following a carotenoid-rich diet for one week, CTD blood levels rose to their original value. This is because the human body can absorb large amounts of carotenoids from natural foods (e.g., vegetables and fruits).

So far, over 700 CTDs have been identified and researched; however, only a small number of these phytochemicals have been found in human blood and tissues (about 24 nutrient chemicals) [76,215]. Patients with HNC and oral leukoplakia have generally been found to have lower plasma concentrations of β-carotene, lycopene, and other CTDs compared with healthy people [216,217]. For instance, Sakhi et al. [218] noted that HNC patients exhibited significantly increased concentrations of intracellular factors related to oxidative stress before treatment with radiation therapy compared with healthy people and that they demonstrated relatively higher levels of reactive oxygen species in plasma, an indicator of oxidative stress degree. It was also confirmed that substantially higher plasma levels of CTDs after radiotherapy were present in HNC individuals. In addition, preradiotherapy plasma concentration of CTDs and other antioxidant agents was associated with overall survival in HNSCC patients. Among the analysed biomarkers, high blood CTD concentrations before radiotherapy were noted to be linked to progression-free survival during prolonged follow-up (HR = 0.42, 95% CI: 0.20–0.91, *p* = 0.03). Biomarkers of antioxidants and oxidative stress, such as carotenoids, were negatively related to prognosis in HNSCC subjects treated with radiotherapy compared with healthy controls. Increased activity of antioxidant agents prior to radiotherapy and greater oxidative stress during radiotherapy was associated with improved overall survival, indicating that in HNSCC patients, various oxidative molecules and pathways with oxidative properties are responsible for the determination of survival before and during this type of management. The authors underline the importance of optimizing antioxidant status and oxidative stress in treating these patients.

Most importantly, the side effects resulting from radiotherapy have a significant impact on lifestyle, such as eating. Consequently, many HNC patients report lower fruit and vegetable intake, which lessens antioxidant intake; this is compounded by free radical (ROS) formation resulting from the radiation therapy, resulting in lower blood antioxidant levels. Because of the location of tumours in the head and neck region and the unfavourable symptoms of the application of radiotherapeutic procedures, the mean level of β-carotene and other CTD dietary consumption among the HNC population is very often 50% less than that of healthy people [82]. For example, Sakhi et al. [82] report lower concentrations of carotenes, lycopene, lutein, zeaxanthin, and total CTDs in the plasma of subjects with HNC postradiotherapy compared with healthy controls. This may be due to the observation that HNSCC patients have a much lower average intake of fruits and vegetables containing CTDs and are a group at risk of higher incidence of oral, pharyngeal, and laryngeal cancer. Moreover, scientists indicate that an increase in plasma CTD levels after radiotherapy may significantly reduce the risk of shortened survival or the presence of a relapse in the HNC population, and the concentration of CTDs in the blood may increase with greater consumption of carotenoid-rich foods.

A number of researchers have also highlighted that supplementation of β-carotene in high doses or consumption of CTD-rich foods are good clinical practices that may reduce the incidence and frequency of local and systemic complications after radiotherapy [219]. This has been attributed to the protective influence of β-carotene on normal tissues against radiation injury. For instance, a study of 540 HNSCC patients found that those with a low average consumption of carotenes during a treatment course of radiotherapy were at a higher risk of severe complications linked to management and tumour relapses [219]. Higher β-carotene dietary supplementation was found to be significantly related to fewer severe acute side effects (OR = 0.61, 95% CI: 0.40–0.93), and similar effects were observed for serum β-carotene (OR = 0.73, 95% CI: 0.48–1.11). Subjects with a higher concentration of β-carotene also had a lower frequency of regional tumour relapses (HR = 0.67; 95% CI: 0.45–0.99). However, it should be noted that CTD supplements were found to be connected with a higher recurrence rate; this phenomenon was possibly due to the antioxidant capacity of these phytochemicals that also protects neoplastic cells from radiation treatment consequences, thereby negatively interfering with radiotherapy. By contrast, the β-carotene derived from healthy food rich in CTDs poses no such problem.

Sakhi et al. [82] also observed that HNC individuals exhibited 3 to 4-fold decreased CTD levels after receiving radiation therapy than a reference control group. Longitudinal research on overall survival in HNC patients confirmed that higher postradiation carotenoid, i.e., carotenes and lutein, levels in plasma were significantly correlated with longer progression-free survival, highlighting the protective role of these compounds.

Other studies indicate that antioxidant and anti-inflammatory nutritive phytochemicals, such as CTDs, may moderate a final activity of proinflammatory lymphokines, reducing the frequency of metastasis and extending overall survival rate in HNC. For instance, Arthur et al. [220] performed a cross-sectional study of 160 newly diagnosed HNC individuals enrolled in the University of Michigan Head and Neck Specialized Program of Research Excellence (HN-SPORE) who completed pretreatment food semi-quantitative Harvard FFQ and health surveys. The authors observed significant inverse associations between TNF-α, IL-6, and IFN-γ plasma concentration and total CTD consumption (*p* = 0.006, *p* = 0.04, and *p* = 0.04, respectively). Significant, inverse Spearman test correlations were also observed between sample days and β-carotene (ρ = −0.29, *p* = 0.001), α-carotene (ρ = −0.28, *p* = 0.001), and total CTDs (ρ = −0.22, *p* = 0.01). Furthermore, significant negative relationships were also noted if quartiles of lycopene intake for IL-6 were taken into account; however, no significant associations were reported for relationships between IL-6 and TNF-α and plasma carotenoid levels. However, a substantial tendency was also observed towards decreased IFN-γ levels across increasing quartiles of β-cryptoxanthin concentrations. These researchers suggest that other nutritive factors from food that are highly correlated with dietary CTD intake may be behind the relationships between reported carotenoid consumption and proinflammatory interleukin levels.

Consumption of proper amounts of phytochemicals was also shown to determine the physiological response to inflammation and intracellular oxidation processes [36,221,222,223,224]. High scores for whole food dietary pattern and total carotenoid intakes before treatment were significantly linked to lower plasma concentrations of the most important proinflammatory interleukins (i.e., IL-6, TNF-α, and IFN-γ), independent of other cellular agents known to regulate IL levels. Additionally, an increased dietary intake of lycopene was substantially related to decreased IL-6 levels. The researchers proposed a mechanism by which a whole-food diet can influence circulating cytokine levels in individuals with HNC; this may be due to modulation of the kappa B nuclear pathway. A number of epidemiologic and clinical data have clearly indicated that NF-κB may be an important factor in activating the tumour progression, making this signalling a potential target for therapeutic intervention, more so that it has been proven that the above-mentioned interleukins can all stimulate this key pathway. ROS and RNS produced during oxidative stress has also been reported to stimulate NF-κB transcription factor signalling, further upregulating the activity of proinflammatory cytokines [223,224].

In summary, eating habits have been related to the carcinogenesis of various cancers, including HNS. The presence of inflammatory serum factors such as interleukins (i.e., IL-1β, IL-4, IL-6, IL-10, and TNF-α) have been estimated in persons receiving certain types of diet, and the results indicate that consumption of higher total carotenoids reduces this inflammatory condition [225]. The balance between these factors and those with more proinflammatory properties will impact the anti-inflammatory action of the nutrition in these patients. While the biological phenomena that predispose patients to higher inflammatory responses remain largely unknown, epigenetic modulation such as DNA methylation has also been taken into account as a cause of systemic inflammation. For instance, in a pilot randomized clinical trial, Moody and collaborators [36] identified high circulating lycopene levels in one group of HNC survivors with various stages of cancer of the oral cavity, pharynx, or larynx after posttreatment ≥6 months; these patients demonstrated substantial differential epigenetic methylation status of transcriptional and translational regulators, as well as of targeted genes of receptor signalling in the immunocompetent T cell (i.e., hypermethylation of CD40 ligand and Tec protein tyrosine kinase and hypomethylation of CD8A). The researchers also identified another HNC group from a posttreatment cohort with history of alcohol abuse, and these again demonstrated a lower rate of addition of a methyl group on the toll-like receptor (TLR) signalling, i.e., hypomethylation of TLR5, a compound of the inhibitory pathway in nuclear factor-kappa B kinase complex (CHUK) activity and MAP3K8 and MAP2K3 kinases. These findings suggest that high alcohol consumption may be linked to crucial changes in DNA methylation, especially in leukocytes [226,227]. Alcohol abuse may impact folate absorption and disturb one-carbon metabolism, and it has been observed to induce and enhance the hypomethylation process during head and neck carcinogenesis; DNA hypermethylation was confirmed for mitochondrial proteins of immunocompetent white blood cells in HNSCC survivors. Mitochondrial ribosomal proteins in leucocytes are believed to be involved not only in oxidative phosphorylation but also in apoptosis [228,229].

An important study examined the relationship between pretreatment serum carotenoid intake and the prognosis of HNC in 154 newly diagnosed HNC subjects enrolled in the UM HN-SPORE program [230]. In adjusted analysis, subjects with increased blood xanthophyll (including the oxygenated carotenoids lutein, β-cryptoxanthin, and zeaxanthin) levels and total CTD concentrations were characterized by a longer recurrence-free time over a median follow-up time of 37 months than those with low levels (*p* = 0.002 and *p* = 0.02, respectively). Furthermore, overall survival was significantly better in patients with high blood xanthophyll levels compared with those with low concentrations (*p* = 0.02). Some insignificant trends were observed between increased plasma β-carotene and lycopene levels and decreased recurrence rate. It is unclear whether the protective effects of xanthophyll in cells are due to these CTDs or of some other phytochemicals that are rich in xanthophylls. While α- and β-carotenes are present in a higher amount in fruits and vegetables than the other measured CTDs, lutein and zeaxanthin are present mainly in green leafy vegetables, which are typically rare in the European and US diets; this may be the reason for the discrepancy in the observed results between various populations.

It is also important to emphasize that many research studies have yielded consistent results related to the effects of diets rich in fruits and vegetables and HNC. Increasing daily vegetable and fruit consumption to eight servings for three months was linked to significantly higher CTD concentration in the blood and improved overall survival. In summary, increasing phytochemicals intake containing a large amount of CTDs can therefore prevent the onset and progression of HNC and reduce the frequency and severity of radiotherapy-induced complications and local recurrences in these patients. Chan et al. [231] provide some evidence that the risk of developing HNS increases with carotenoid deficiency. Of a total of 194 subjects with oral cavity malignancy enrolled in the research, 46% had β-carotene deficiency; in addition, of these, more than half of the patients with lower stages of neoplastic lesions (pT0–pT2 stages) had β-carotene deficiency, and nearly 40% of patients in the pT3–pT4 stages had β-carotene deficiency. In this cohort, tumour progression was lined to changes in cellular metabolism, resistance to programmed cell death, and increased production of ROS. It is believed that oxidative stress increased the likelihood of cancer developing and that reactive species stimulated proinflammatory factor secretion to facilitate malignant dissemination and chronic inflammation. In summary, since high oxidative stress and inflammation is often present in HNC individuals, supplementation with antioxidant vitamins such as ubiquinone or β-carotene could be preferentially used. This may counter the higher levels of oxidative stress which can exacerbate cancer development, as well as ROS production, which may also stimulate proinflammatory cytokine secretion, promoting tumour progression and metastasis. A randomized, double-blind, placebo-controlled trial by Datta et al. [173] confirmed that diet supplementation with multiple FV concentrates, i.e., Juice PLUS+, had a substantial impact on p27^Kip−1^ and Ki-67 expression, crucial factors associated with the risk of SPTs. The researchers randomized 134 HNC patients to 12 weeks of JP or placebo. A significantly higher serum concentration of α-carotene (*p* = 0.004), β-carotene (*p* < 0.0001), and lutein (*p* = 0.0004) was noted after 12 weeks of complementary dietary treatment. No significant association was however observed in CTDs (except lycopene; *p* = 0.019) in the placebo reference group.

An important issue is that smoking that may modify the associations between serum nutrient levels and mortality. Mayne et al. [232], in a longitudinal chemoprevention study of 259 participants with HNC, assessed the effect of micronutrients on oral, pharyngeal, or laryngeal SPCs. These were analysed by HPLC for selected micronutrients. All-cause mortality (primary outcome) and cause-specific mortality (secondary outcomes) were measured in relation to plasma micronutrient levels at selected time points. In models adjusted for time-dependent smoking, treatment arm, study site, and gender, only plasma lycopene was substantially inversely linked to total mortality (HR = 0.53, 95% CI: 0.30–0.93, above vs. below median). Tobacco use was observed to modify the association between plasma nutrient levels and patient survival. Lycopene (HR = 0.08, 95% CI: 0.02–0.36), α-carotene (HR = 0.25, 9 5% CI: 0.09–0.73), and total carotenoid levels (HR = 0.22, 95% CI: 0.07–0.70) were inversely associated with survival time in non-smokers. In summary, total serum carotenoid levels were found to have protective effects on mortality during posttreatment of patients with head and neck cancer who were smokers and disease-free at the time of blood sampling.

A key study by Arthur et al. [230] examined the relationship between pretreatment serum CTDs and prognosis in 154 patients newly diagnosed with HNC. As a result, a significantly larger group of individuals with advanced tumour stages (SIII or SIV) was observed to have low serum xanthophyll concentrations. In addition, current smokers were found to be significantly less likely to have the highest serum xanthophyll and serum CTD concentrations. Patients with higher consumption of leafy green vegetables, dark yellow and orange vegetables, and fruit on an FFQ also had significantly higher serum xanthophyll concentrations. Patients with high serum xanthophylls also had significantly higher average survival rate than those with low concentrations. However, no significant trends were noted between higher β-carotene and lycopene concentrations and tumour relapses. In addition, no apparent associations were confirmed between recurrence rate and α-carotene levels or between prognosis and carotenes, lycopene, or total CTD levels. The findings indicate that high pretreatment serum xanthophyll levels were linked to a lower incidence of recurrence and survival time, independent of other factors known to influence these outcomes. Importantly, however, it was observed that the relationship between HNC mortality and CTDs may be affected by smoking consumption. As has been previously reported, it is possible that CTDs show undesirable effects when consumed in high doses from supplements, especially in heavy smokers. Xanthophylls may demonstrate greater heterogeneity with regard to dietary intake and blood concentration than other carotenoids due to being present in smaller amounts in a vegetarian diet.

Another randomized clinical trial performed in 540 subjects with HNC by Meyer et al. [219,233] also confirmed that smokers who supplemented their diets with β-carotene had a higher tumour relapse rate, mortality, and SPC incidence compared with non-smokers and the reference placebo group. Interestingly, however, higher reported β-carotene consumption, i.e., not from supplements, before research was linked to a lower incidence of local recurrence, regardless of smoking status.

Importantly, the differences reported between these cohorts could also be due to the HPV status of the tumour [234,235]. A number of studies have reported that HPV(+) cancers were related to longer survival time compared with the HPV(−) cohort; however, HPV-positivity was also related to higher intakes of chemoprotective micronutrients. One limitation of the study was its relatively small sample size and incomplete data on posttreatment smoking status for 30.6% of the study population. Another limitation was that the researchers could not evaluate subjects by type of anticancer treatment received. In cases with the most aggressive stage (SIV), i.e., the majority of the cohort studied, the efficacy of chemotherapy and radiation management was difficult to estimate. The scientists also observed a significant, negative association between sample storage time and CTD levels, suggesting that there may have been some degradation of these nutrients in the older serum samples used in the analysis.

Regardless of the differences between their findings, it is worth noting that these studies provide an insight into the inflammatory and epigenetic changes linked to lifestyle in HNSCC survivors; these undoubtedly affect tumour recurrence, patient survival, and quality of life in HNC survivors.

Interventional studies are necessary for determining the real benefits of dietary supplements or dietary CTDs in the treatment and chemoprevention of HNC. Only two such studies could be found that had both randomisation and a double-blind design. Crowder et al. [37] performed an interventional study that was a two-arm randomized observational clinical trial (RTC) among 24 posttreatment HNC survivors with the aim of evaluating improvements in survival and cancer-related biomarkers, such as lycopene, lutein, zeaxanthin, β-carotene, and cryptoxanthin. The survivors were categorised according to epigenetic changes, i.e., DNA methylation of functionally important molecules in leukocytes, and the relationship between the addition of a methyl group on a substrate and inflammatory parameters was determined with regard to circulating CTDs and cytokine levels. The experimental group received weekly 15–30 min telephone dietary counselling from a RDN stressing 2.5 cups/week of cruciferous vegetables (CV) and 3.5 cups/week of green leafy vegetables (GLV), with emphasis on their importance as a source of carotenoids and proinflammatory factors; the attention control group received 15–30 min telephone dietary counselling each week focusing on general healthy eating for cancer survivors. The protocol was approved by the American Institute for Cancer Research and Neck Specialized Program of Research Excellence (HN-SPORE). The carotenoid analysis was performed as part of the NIST Micronutrients Measurement Quality Assurance Program.

No substantial differences were noted between the analysed groups with regard to changes in circulating pre- and post-intervention interleukins, such as IFN-γ, IL-1β, IL-6, and TNF-α following the intervention. In addition, no substantial differences in CTD serum levels (i.e., β-carotene, lycopene, lutein, zeaxanthin, cryptoxanthin, xanthophylls, or total CTD) were observed between pre- and post-intervention tests for either arm; furthermore, at the participant level, no differences were observed between the concentration of CTDs in sera and CV/GLV intake before and after the intervention. However, the research was performed in a relatively short period (12 weeks) and in a small group [37].

In addition, no proven similarities were confirmed between the two populations regarding mean change in total serum carotenoid concentrations after the intervention. Relatively large differences (0.2 to 0.8), except β-carotene and xanthophylls, suggested a non-trivial difference between groups regarding an average shift in lycopene, lutein, zeaxanthin, cryptoxanthin, and total carotenoid levels. Although not statistically significant, the results suggest that the intervention resulted in higher circulating cytokine and carotenoid levels. The discrepancy of effects—i.e., 0.3–0.5 for all cytokines and 0.2–0.8 for the carotenoids lycopene, lutein, zeaxanthin, and cryptoxanthin and total CTD level—point to a possible difference in mean change [37].

Comparable observations were noted by Cartmel et al. [236] in a randomized clinical trial of curatively treated early-stage HNC patients (clinical stage I and II); the researchers confirmed higher levels of plasma CTDs in the research group than in the reference group, although no significant differences in serum carotenoid levels were found between these populations. The authors of two RCT studies emphasized the importance of green leafy vegetable consumption in lowering inflammatory cytokine levels; however, they indicated that more extensive RTC studies are needed to evaluate the effectiveness of this intervention on patient outcome and prognosis in HNC. They also highlighted the fact that up to 47% of HNC patients have a problem with dietary intake due to essential local disturbances, e.g., xerostomia, dysphagia, and taste alternation, which must affect the final conclusions obtained [237,238].

Finally, it is also worth recalling the importance of ATRA (all-trans retinoic acid) in the chemopreventive cellular effects and advanced therapy in head and neck cancers of various origins. ATRA also stays an important alternative for the inhibition of disease progression and further advancement in many types of human tumours [239,240,241,242,243,244]. ATRA has been found to regulate cell division and inhibit the G1 phase of the cell cycle, most likely by modifying gene transcription. This inhibition may be associated with, inter alia, a lower action of cyclin D1 and Bcl-2 and a higher function of p27^Kip−1^ and Bcl-xL or Bax, a decrease in the hyperphosphorylation of pRB, and the proper action of JNK1 [239,242]. Interestingly, the constitutive activation of EGFR and STAT3 pathways in HNC cells was more commonly associated with resistance to treatments with ATRA [239,241,242]. Scientific studies indicated that nuclear retinoic acid receptors (RARs) may function as the mediators of ATRA and thus may act as factors affecting gene expression in tumour cells. Recent research indicates that retinoids inhibit the development of neoplastic changes in the head and neck and reduce their incidence [242]. A selective loss of RARβ expression was observed in the initial lesions, which researchers associated with the promotion of further stages of carcinogenesis [242,243]. ATRA regulates cell differentiation and inhibits the G0 cell cycle phase that is associated with activity of the MAPK pathway, translocation of c-Raf to the nucleus, and activation of the Src family kinases and PI3K [224].

In conclusion, many multicentre studies have also shown a significant improvement in the rates of morbidity, mortality, and quality of life through dietary modification in patients with HNC. It also appears justified to register the presence of CTDs in foods by using proper dietary protocols into HNC survivorship care plans. A larger population of studies is warranted to estimate the effectiveness of this dietary intervention on malignant initiation, progression, and clinical outcomes.

## 4. Conclusions

The effect of diet and nutrition not only on health but also on the initiation, development, and advancement of HNC is well-documented. However, due to the complex nature of carcinogenesis and the multifactorial influences and interactions between dietary phytochemicals and well-established risk factors, i.e., tobacco and alcohol abuse, it is difficult to unequivocally define the importance of carotenoids in the aetiopathogenesis of HNCs of various origin. Although recent publications indicate that relationships exist between dietary carotenoid serum/tissue levels and the stages carcinogenesis, as confirmed in vitro and in animal models, their findings do not allow a clear and unambiguous association to be made between the multidirectional effects of carotenoids and the aetiology of HNC. Nevertheless, a higher level of dietary CTDs, such as carotenes, astaxanthin, fucoxanthin, β-cryptoxanthin, and lycopene, and consequently a higher concentration in blood serum, positively influences the inhibition of the molecular landscapes of carcinogenesis and tumour progression; this is believed to take place by the repression of key intracellular signalling processes, i.e., NF-κB, AKT, ERK, and subfamilies of MAPKs and by inhibiting the PI3K/AKT/mTOR pathway and cell cycle regulation at critical points. These pathways also appear to inhibit proliferation and angiogenesis and to influence GJIC and MDR phenomenon, thus reducing the enhancing efficacy of chemotherapy against HNC and other cancers. Moreover, dietary carotenoids and other phytochemicals may act as antioxidants, anti-inflammatories, immunoprotectors, immune regulators, cellular membrane stabilizers, and oncogenic signalling and apoptosis regulators, as well as cell cycle and angiogenesis controllers. In particular, the scavenging ability of carotenoids reduces reactive oxygen species (ROS), favours DNA repair, negatively regulates oncogenic transcription, and stimulates a number of key genes encoding antioxidant enzymes.

Many multicentre studies suggest that dietary carotenoids may have potential preventive and treatment roles in cancer; however, a number of studies have inconclusive final results, which may be due to short observation periods or variable follow-up times, large diversity of the studied populations, and often too small groups of cases and controls. Nevertheless, other studies have indicated that under exceptional conditions characterised by unbalanced intracellular redox status and high oxidant status, high carotenoid concentration can act as an inducer of prooxidant reactions. Despite the above observations, it should be emphasized that most recent publications clearly indicate that a proper content of consumed CTDs may inhibit the process of initiation, progression, and metastasis of cancer. A diet containing CTDs provides many advantages and potentially allows the regulation of HNC development by repressing initiation, promotion, and progression events. Further studies with long-term follow-ups, larger study groups, and prolonged observations of the use of CTDs in case–control studies are necessary before unequivocal results can be shown regarding the role of dietary carotenoids in carcinogenesis. Currently, it seems that the WHO recommendations emphasising the value of consuming foods rich in CTDs is justified. Furthermore, a proper BMI, physical activity, and the avoidance of excessive smoking and alcohol abuse also remain essential factors in a healthy lifestyle.

## Figures and Tables

**Figure 1 nutrients-14-00531-f001:**
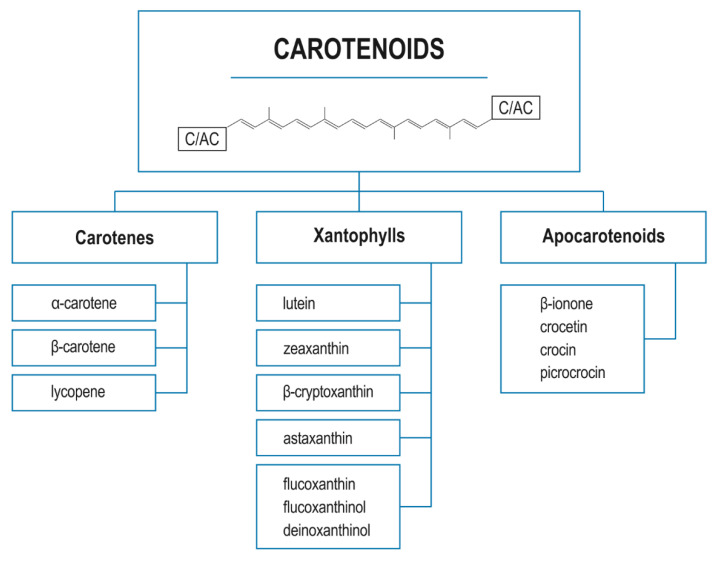
A classification of the major representatives of the carotenoids, viz. carotenoids, xanthophylls, and apocarotenoids. C/AC: cyclic/acyclic end groups differ in individual carotenoids.

**Figure 2 nutrients-14-00531-f002:**
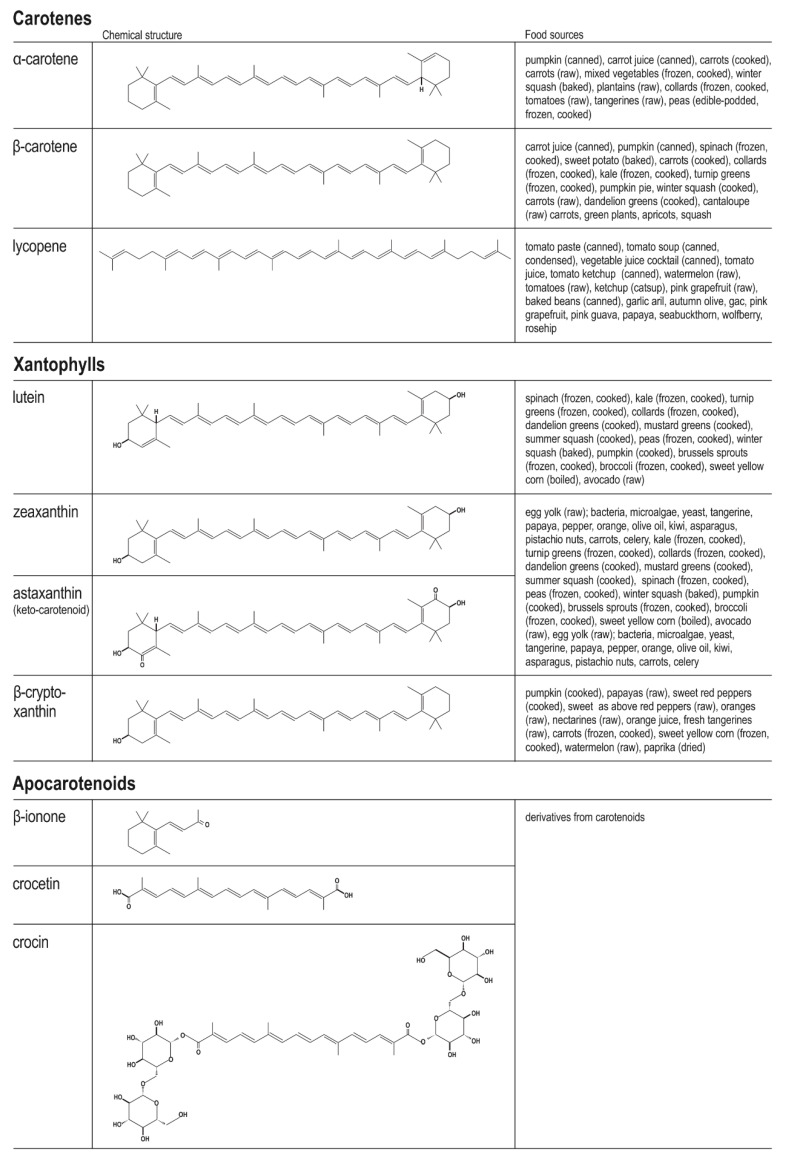
Examples of the major dietary carotenoids, their chemical structure, and food sources.

**Figure 3 nutrients-14-00531-f003:**
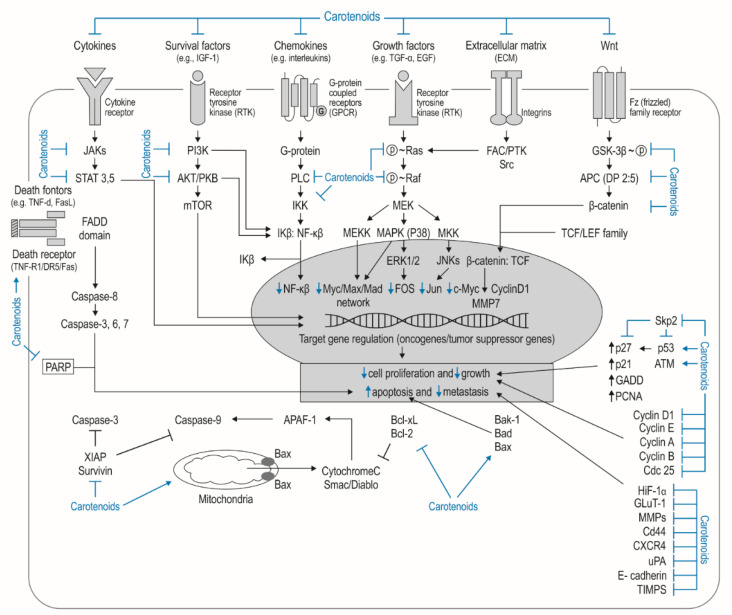
Some potential biological intracellular targets of carotenoids in head and neck cancer (HNC). JAK: Janus kinases, STAT: signal transducer and activator of transcription proteins, FADD: FAS-associating death domain-containing protein, PARP: poly (ADP-ribose) polymerase, XIAP: the X-linked inhibitor of apoptosis protein, NF-κB: nuclear factor kappa-light-chain-enhancer of activated B cells, IκB: IkappaB kinase or IKK, PLC: phospholipase C, APAF-1: apoptotic protease activating factor 1, Smac/Diablo: second mitochondria-derived activator of caspases, PI3K: phosphoinositide 3-kinases (or phosphatidylinositol 3-kinase), AKT/PKB: protein kinase B, mTOR: the mammalian target of rapamycin, MAPK/MKK/MEKK: mitogen-activated protein kinase kinases, EGF: epidermal growth factor, Ras/Raf: RAF proto-oncogene serine/threonine-protein kinases, JKN: c-Jun N-terminal kinase, MMP: matrix metallopeptidase or matrixin, Src: proto-oncogene tyrosine-protein kinase Src, GSK3β: glycogen synthase kinase 3 beta, APC (DP 2.5): polyposis coli (APC) or deleted in polyposis 2.5, TCF/LEF: T cell factor/lymphoid enhancer factor family, Bcl-xL: B-cell lymphoma-extra-large, Bcl-2: B-cell lymphoma 2, Bak-1: Bcl-2 homologous antagonist killer, Bad: the Bcl-2-associated agonist of cell death, Bax: apoptosis regulator or Bcl-2-like protein 4, Skp-2: S-phase kinase-associated protein 2, p53: tumour suppressor p53, ATM: ATM serine/threonine kinase checkpoints, the G1/S and the G2/M, during the cell cycle, GADD: growth arrest and DNA-damage-inducible protein GADD45 gamma, PCNA: proliferating cell nuclear antigen, p27^Kip1^: cyclin-dependent kinase inhibitor 1B, p21^Cip1/Waf1^: cyclin-dependent kinase inhibitor 1 or CDK-interacting protein 1, HIF-1α: hypoxia-inducible factors alpha, GLUT-1: glucose transporter 1, CXCR4: C-X-C chemokine receptor type 4, uPA: urokinase-type plasminogen activator, TIMPS: tissue inhibitors of metalloproteinases (TIMPs); ~P: phosphorylated form.

**Figure 4 nutrients-14-00531-f004:**
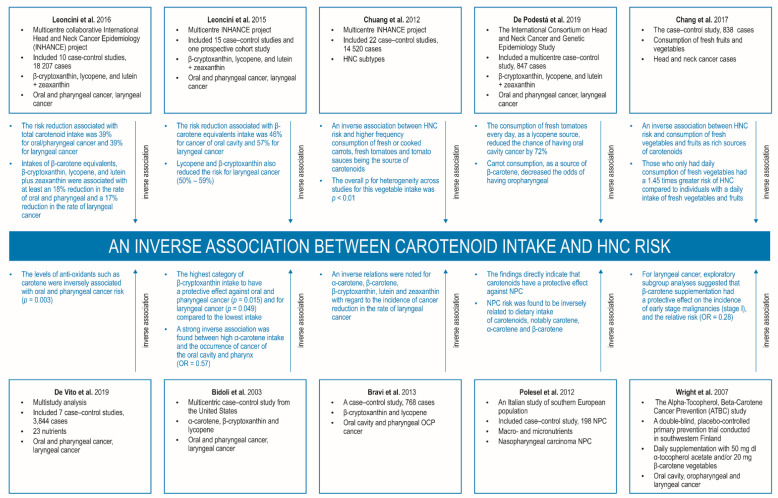
The representative scientific publications confirming an inverse association between carotenoid intake and HNC risk [15,16,17,19,180,182,183,185,186,187].

**Figure 5 nutrients-14-00531-f005:**
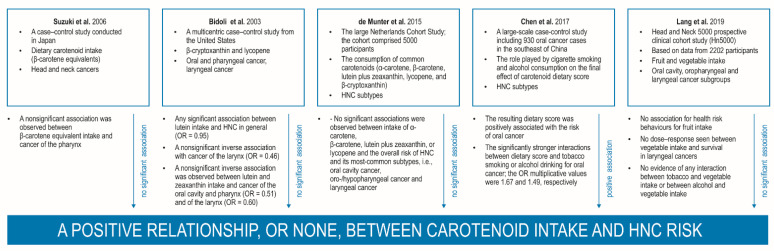
The representative scientific articles confirming a positive relationship, or none, between carotenoid intake and HNC risk [185,189,191,192,199].

## Data Availability

Data available in a publicly accessible repository.

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
