# Peer review of "Dietary Carotenoids in Head and Neck Cancer—Molecular and Clinical Implications"

_nutrients, 2022, doi:10.3390/nu14030531_

Round 1
Reviewer 1 Report
The article by Starska-Kowarska on Dietary Carotenoids in Head and Neck Cancer is well written. From my point of view, it would be useful to replace some descriptive parts with explanatory Figures. Overall, I consider the article to be thoroughly described and in particular I recommend the author to check the text formatting.
I have only minor comments:
Starska-Kowarska Katarzyna 1,2*Prof., MD and PhD ? I am convinced that academic degrees are not necessary.
Were the Figures drawn by author? If not copyright permission is required.
page 20 L884-898 - this part should be checked and formated (N,N,N,N; 12-O = should be in italics).
references 184 and 185 - check formating
Reviewer 2 Report
Thank you, authors, for the opportunity to review this interesting work. Carotenoids have been demonstrated as potential inhibitors of cancer cells and your work quite nicely summarizes the body of literature present on this topic. However, you refrain from placing your opinion in the manuscript. Citing references 26-31, you do mention that there are certain publications that do not confirm the observation that carotenoids or their serum concentrations are not associated with onset, progression and metastases in cancers of various origin. Please explain in your understanding what could be the shortcomings of this work or why this work does not dilute the other evidence that carotenoids have good anticancer activity?
Also- Figure 3: it is a beautiful image very neatly made. Can I ask which software did you use to design it so that I can also make such neat figures. I do have some recommendations to improve the image. In one instace you show that carotenoids (CTDs) increase Caspase-3,6,7 with a straight arrow. In another instance no connection between caspase-3 and CTDs is shown (the XIAP, Survivin pathway), why? In my understanding the first instance where there is a solid arrow should not be present as CTDs do not lead to the activation or phosphorylation of Caspases, rather it activates the Death receptors which then lead to the downstream activation of caspases. Please see if this is the case then modify the image. In similar fashion, the CTDs inhibit phosphorylation of GSK3b by Fz and similarly the downstream activation. Please show that by placing the inhibitor signal closer to the arrows. In similar manner, do CTDs inhibit Ras or their phosphorylation?
Add more information about ATRA in the literature.
Round 2
Reviewer 2 Report
Thank you for considering the suggestions and answering the queries to my satisfaction. Hope this work helps re-ignite the interest in carotenoids as anticancer compounds. I do not have any further comments/queries/suggestions.